# Characterisation of the manchette architecture and its role as transport scaffold using cryo-electron tomography

Jo H Judernatz[1] , Laura Pérez Pañeda[2] , Tereza Kadavá[2] , Albert JR Heck[2] , Tzviya Zeev-Ben-Mordehai[1]

The development of correctly shaped sperm cells is crucial for male reproductive health and fertility. The manchette is a transient microtubule-based structure that assembles during spermiogenesis and contributes to sperm head shaping. Defects in the manchette can cause sperm deformations and subsequent infertility. Previous studies have suggested that the manchette acts as a cellular transport platform, distributing proteins and vesicles during spermiogenesis in a process known as intra-manchette transport. The manchette and intra-manchette transport are still poorly understood, as high-resolution imaging is missing. Here, we used cryo-electron tomography and proteomics to visualize the manchette and identify some of its transport components. We characterize the overall architecture of the manchette and show that its perinuclear ring thickens as the structure constricts. We observed for the first time dynein directly interacting with the manchette. We further find F-actin as single filaments and filament clusters intercalating with the manchette microtubules. Our results provide new insights into the manchette's architecture and potential role as a transport scaffold, highlighting its significance for the shaping of sperm cells during spermiogenesis.

## Introduction

Spermiogenesis is the post-meiotic process in which round spermatids are morphologically reorganized into motile sperm cells (O'Donnell, 2014). With the onset of nuclear condensation, the manchette assembles as a temporary microtubular scaffold around the distal half of the spermatid's nucleus to aid in sperm head shaping (Rattner & Brinkley, 1972; Kierszenbaum, 2001; Lehti & Sironen, 2016). The manchette is composed of thousands of microtubules radiating from a dense perinuclear ring (Fawcett et al, 1971; Rattner & Brinkley, 1972). Throughout spermiogenesis, the manchette gradually migrates down the spermatid nucleus, while simultaneously constricting to shape the nucleus (Russell et al, 1991; Lehti & Sironen, 2016). By the end of nuclear condensation, its length reaches a maximum and is subsequently disassembles (Clermont et al, 1993; Dunleavy et al, 2019a).

In addition to its active involvement in sperm head shaping, the manchette appears to provide a cytoskeletal platform for bidirectional transport in the spermatid required for the significant reorganization of cellular material to form the highly polarized sperm cell (Fawcett et al, 1971; Kierszenbaum et al, 2003b; Kierszenbaum & Tres, 2004). Active transport of components along the manchette was termed intra-manchette transport (IMT) (Yoshida et al, 1994; Kierszenbaum, 2002; Kierszenbaum et al, 2003b; Hayasaka et al, 2008; Lehti et al, 2013). IMT has been proposed to ensure the timely delivery of macromolecules, vesicles, and mitochondria to the developing axoneme of the sperm tail, involving multiprotein complexes (Kierszenbaum, 2001, 2002; Kierszenbaum et al, 2003b; Lehti & Sironen, 2016; Yu et al, 2020), and to redistribute nuclear material like protamines (Agudo-Rios et al, 2023). It was suggested that IMT depends on the cytoplasmic microtubule (MT) motor proteins dynein and kinesin (Yoshida et al, 1994; Kierszenbaum, 2002; Li et al, 2016; Agudo-Rios et al, 2023). Cytoplasmic dynein requires the binding of dynactin and adaptor proteins to activate and tether cargo (Torisawa et al, 2014; Xiang & Qiu, 2020). Once bound to dynactin, the motor domain of dyneins can undergo a mechanochemical cycle to move cargo along a microtubule in a plus-end to minus-end direction (Gennerich & Vale, 2009; Canty et al, 2021). Most kinesins, on the other hand, move along microtubules in a minus-end to plus-end direction with few exceptions (Noda et al, 2001; Ogren et al, 2022). Kinesins are involved in many molecular processes besides cargo transport along MTs (Ali & Yang, 2020). Although kinesins, dyneins, and dynactin subunit p150[glued] were shown to localize to the manchette with immunofluorescence microscopy and immunogold labelling, direct visualization of intact complexes on the manchette is still lacking (Yoshida et al, 1994; Saade et al, 2007; Kierszenbaum et al, 2011a).

[1]Structural Biochemistry, Bijvoet Centre for Biomolecular Research, Utrecht University, Utrecht, The Netherlands   [2]Biomolecular Mass Spectrometry and Proteomics, Bijvoet Centre for Biomolecular Research and Utrecht Institute for Pharmaceutical Sciences, Utrecht University, Utrecht, The Netherlands

Correspondence: z.zeev@uu.nl
Jo H Judernatz's present address is Centre for Genomic Regulation (CRG), Barcelona Institute of Science and Technology (BIST), Barcelona, Spain

 

Several other non-motor proteins have been suggested as molecular components of the manchette and are believed to participate in IMT through direct interaction with tubulins or with dyneins and kinesins, respectively (Chen et al, 2016; Teves et al, 2020; Teves & Roldan, 2022). Subunits of intra-flagellar transport (IFT) complexes A and B were amongst the first proteins hypothesized to play a role in IMT, and all IFT genes are expressed during spermatogenesis (Lehti & Sironen, 2016). IFT complexes are crucial for axoneme assembly and maintenance in cilia and form "trains" of repeating kinesin/IFT/dynein-2 repeats moving along the developing axoneme (Lacey et al, 2023). IFT trains are essential to overcome the unidirectionality of MTs in the axoneme. Like the axoneme, the manchette MTs are unidirectionally organized, with their plus ends facing the perinuclear ring (Mochida et al, 1998; Kato et al, 2004). However, the impact of MT unidirectionality on IMT has not been investigated in detail. The involvement of IFT complex B subunit IFT88 in IMT has been suggested (Kierszenbaum, 2002; Kierszenbaum et al, 2011b). *IFT88* mutant mice show defects in sperm head shaping and tail formation (Kierszenbaum et al, 2011b). In addition, IFT20 (Yap et al, 2022), IFT140 (Zhang et al, 2018), and IFT27 (Zhang et al, 2017) were shown to localize to the spermatid manchette. Consequently, similarities between IMT and IFT have been hypothesized (Kierszenbaum, 2002). However, whether IFT trains assemble in the manchette and are involved in IMT remains unclear.

Although dynein- and kinesin-mediated IMT along manchette MTs was proposed for long distances, short-distance transport and accurate targeting have been proposed to be carried out along actin filaments (Goode et al, 2000; Kierszenbaum et al, 2003b, 2011a; Hayasaka et al, 2008). Actin exists as single subunits in a globular form (G-actin) and as polymerized filaments (F-actin). F-actin motor protein myosin Va localizes to the manchette (Kierszenbaum et al, 2003b; Hayasaka et al, 2008), and actin was detected in Western blots of isolated manchettes (Mochida et al, 1998). However, the presence of F-actin in the manchette remained elusive because phalloidin only faintly stains the manchette (Halenda et al, 1987; Kierszenbaum et al, 2003b; Tachibana et al, 2005; Ferrer et al, 2023).

Our current understanding of the manchette and IMT is limited, as neither has been imaged at high resolution in a close-to-native state. Here, we employed cryo-electron tomography (cryo-ET) and proteomics to visualize the manchette in 3D and identify its transport components. Our findings reveal the architecture of the manchette in the absence of fixatives, and show vesicles and dynein–dynactin complexes associated with the manchette microtubules, which sheds light on its function as a transport platform. We further confirm the presence of actin filaments in the manchette and find filaments organized in two distinct ways: as clusters running parallel to MTs or as single filaments. Our study provides significant insight into the role and function of the manchette during sperm development.

# Results

### In situ imaging of the manchette

To study the architecture of the manchette in developing spermatids, we isolated and enriched elongating spermatid fractions on a BSA gradient and used focused ion beam (FIB) milling to generate thin lamellae (Fig 1A). Tomograms collected close to the nucleus showed an ~200- to 250-nm-wide array of parallel microtubules (MTs) (Fig 1B), agreeing with a previous classical EM study (Rattner & Brinkley, 1972). The MTs extended to the perinuclear ring, which marks the upper border of the manchette (Fig 1B). Vesicles and mitochondria colocalized with the manchette (Fig 1C), as was also previously shown in classical EM studies (Kierszenbaum et al, 2003a). Interestingly, aside from MTs, we observed additional filaments close to the nucleus and the manchette (Fig 1D). The highly crowded molecular environment did not allow us to resolve distinct connections between vesicles and MTs.

### The thickness of the perinuclear ring correlates with the manchette length

To systematically investigate the manchette at higher resolution, we isolated manchettes from rat testes in the presence of the MT-stabilizing agent Taxol based on a published protocol (Mochida et al, 1998) and analysed them with cryo-ET. Because of the missing nucleus, the isolated manchettes flattened on the cryo-EM grid, allowing for direct imaging without the need for sample thinning with FIB milling. Isolated manchettes appeared as trapezoids of different dimensions at low-magnification projection images, with the perinuclear ring on one side and the microtubular sheath connected to it (Fig 2A–C). Isolation from whole testes allowed us to capture manchettes from spermatids of different developmental stages, identified by the length and diameter of the manchettes (Fig 2A–C). Measuring the perinuclear ring thicknesses from tomograms revealed high variability, ranging from 100 to 600 nm, with an average of ~250 nm (Fig 2D–F), which agrees with published values and confirms the integrity of the manchette isolated (Rattner & Brinkley, 1972; Mochida et al, 1998). Plotting the PNR thickness against the diameter and length of the manchette further showed a correlation between increasing perinuclear ring thickness and elongation and constriction of the manchette (Fig 2G).

Tomograms of the region of the perinuclear ring showed the high density of the MT sheath (Fig 3A and B), whereas the perinuclear ring appeared as amorphous protein density. The perinuclear rings often remained attached to a membrane (Figs 2 and 3). The membrane did not always remain intact (Fig S1A and B, white arrows). Measuring the distance between the inner membrane leaflet and the upper edge of the perinuclear ring in a population of manchettes with continuous membrane showed a relatively consistent distance of 12–18 nm (Fig 3C). To check whether proteins are involved in maintaining this consistent distance, we collected higher magnification tomograms, which revealed pear-shaped protein densities at the upper border of the perinuclear ring (Fig 3D). The densities had dimensions of about 10 nm × 6 nm with a regular 8-nm spacing. In orthogonal slices of the edge of the PNR, we observed ladder-like repeats with 8-nm spacing (Fig S2A and B), likely different views of the same proteins. Because of the thickness of the PNR, subtomogram averaging was unsuccessful; thus, the molecular composition of these densities remains unknown. Nonetheless, the prevalence of this protein, the

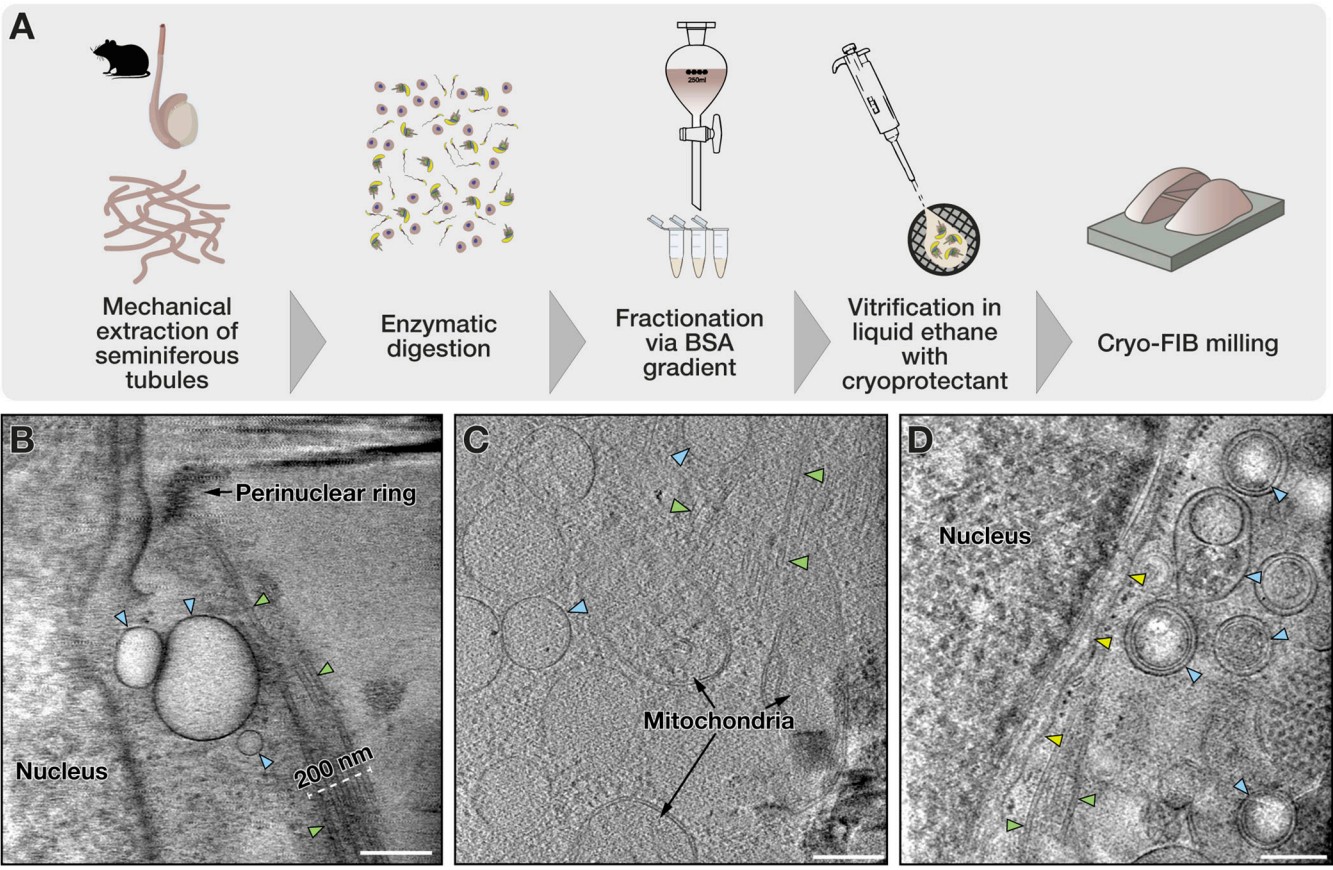

**Figure 1. Vesicles colocalize with the manchette in intact spermatids.**
**(A)** Workflow showing preparation of cryo-FIB–milled lamellae from mouse spermatids. **(B)** A tomographic slice showing the manchette as a 200-nm-wide array of MTs (green arrowheads) with its perinuclear ring close to the nucleus. Vesicles are seen close to MTs (blue arrowheads). **(C)** Tomographic slice showing mitochondria and vesicles colocalizing with MTs. **(D)** A tomographic slice revealing several vesicles and a filament (yellow arrowhead) between the manchette and the nucleus. Scale bars, 250 nm.

ordered arrays it forms, and the uniformity of distance it maintains between the perinuclear ring and membrane all indicate its possible involvement in tethering the membrane to the perinuclear ring.

## Dynein remains bound to isolated manchettes

Like in the in situ data, many vesicles colocalized with the microtubules in isolated manchettes (Fig 4A). Slices from tomograms revealed the very dense packing of the MTs with vesicles of various sizes and shapes between them (Fig 4B and C). To investigate the interaction of the vesicles with the MTs in more detail, we imaged the periphery of the manchette, as these regions were thinner and less crowded. Intriguingly, we observed densities of different sizes and shapes that bridged vesicles to the MTs (Fig 4D–F). However, the densities connecting the vesicles to the MTs were not regular enough for further structural analysis using subtomogram averaging.

Aside from the irregular connections between vesicles and MTs, we often observed prominent densities appearing as hollow rings with slender stalks on the MTs (Fig 5A–C). 3D reconstruction of the densities revealed the dimensions and shape of a cytoplasmic

dynein-1 motor domain (Figs 5D and S3A). The motor domains of cytoplasmic dynein-1 function as dimers and tetramers (Chowdhury et al, 2015; Grotjahn et al, 2018). Consistently, the dynein motor domains on the manchette MTs we observed mainly appeared in pairs or larger assemblies (Fig 5A–C). The dynein motor domains were often trailed by large densities resembling projections of dynein–dynactin complexes (Fig 5E). In agreement, most components of cytosolic dynein–dynactin complexes were detected in our proteomics data of isolated rat manchettes (Fig 5F, Tables S1 and S2). Two exceptions were dynactin subunit p62 (DCTN4) and ARP11 (Chowdhury et al, 2015; Urnavicius et al, 2015). The absence of these common dynactin components in the manchette suggests the possibility of alternative proteins involved in dynein–dynactin complex formation in spermatids. Still, it might also indicate a loss of these proteins during the isolation of the manchette from the spermatid. Cargo is bound to the dynein–dynactin complexes via activating adaptor proteins. Even though we were not able to assign cargo bound to the dynein–dynactin complexes observed in our tomograms, the adaptors RAB11FIP1, HOOK2, and the MT-binding NUMA1 were present in the proteomics data (Tables S1 and S2), suggesting their role in cargo transport in the manchette. The well-characterized

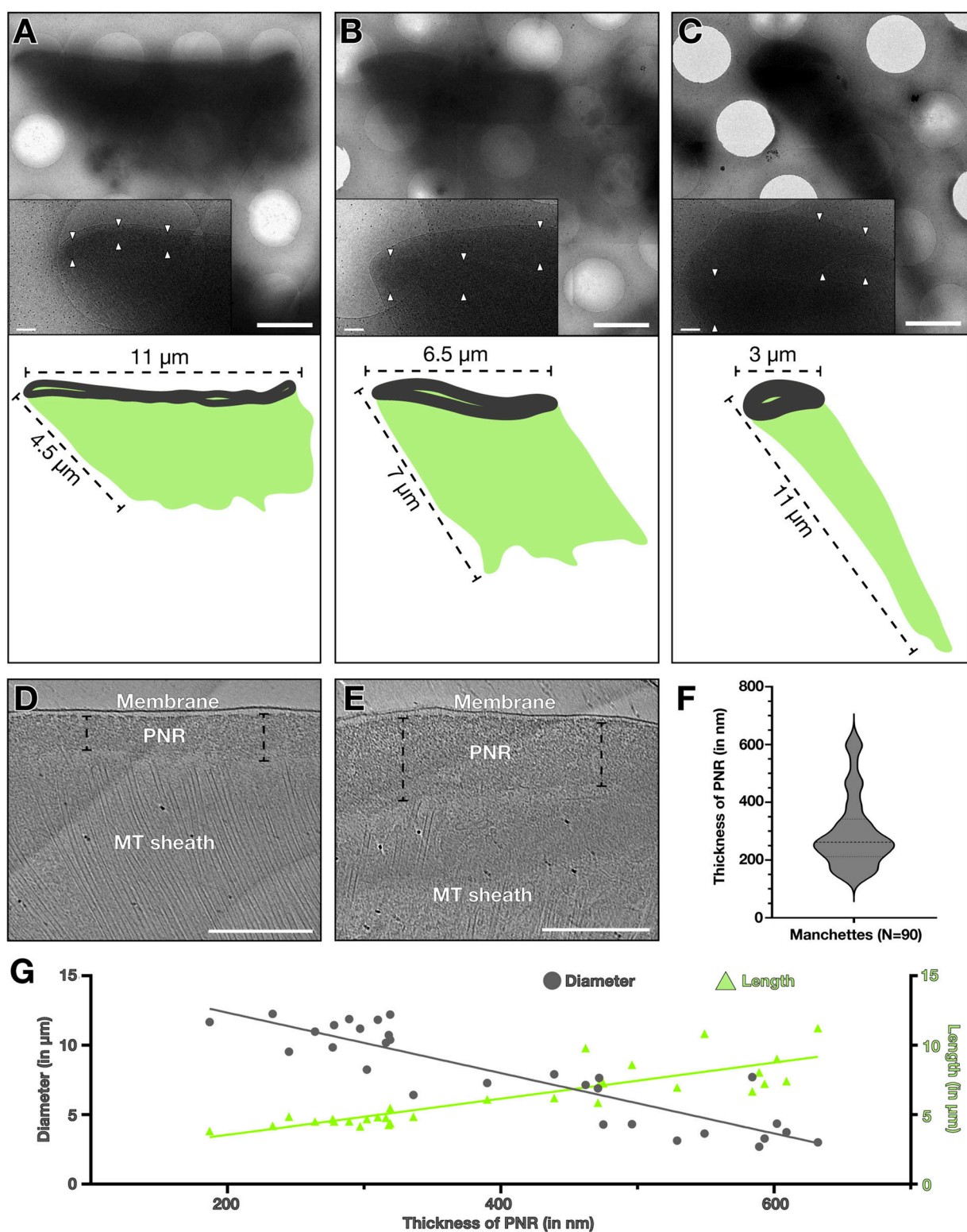

**Figure 2. Increasing thickness of the perinuclear ring correlates to manchette elongation and constriction.**
**(A, B, C)** Projection images of rat manchettes with different dimensions on a cryo-EM grid and annotations below (perinuclear ring [PNR], dark grey; microtubule [MT] sheath, green). Scale bar, 2 μm. The box on the bottom left shows a zoom into the perinuclear ring with white triangles marking its thickness. Scale bar, 250 nm. **(D, E)** Tomographic slices showing the perinuclear ring of different thicknesses. Scale bars, 500 nm. **(F)** Quantification of perinuclear ring thickness across 90 different manchettes. **(G)** Plot of the perinuclear ring thickness relative to the manchette diameter (grey circles) and manchette length (green triangles). Lines show respective linear regressions.

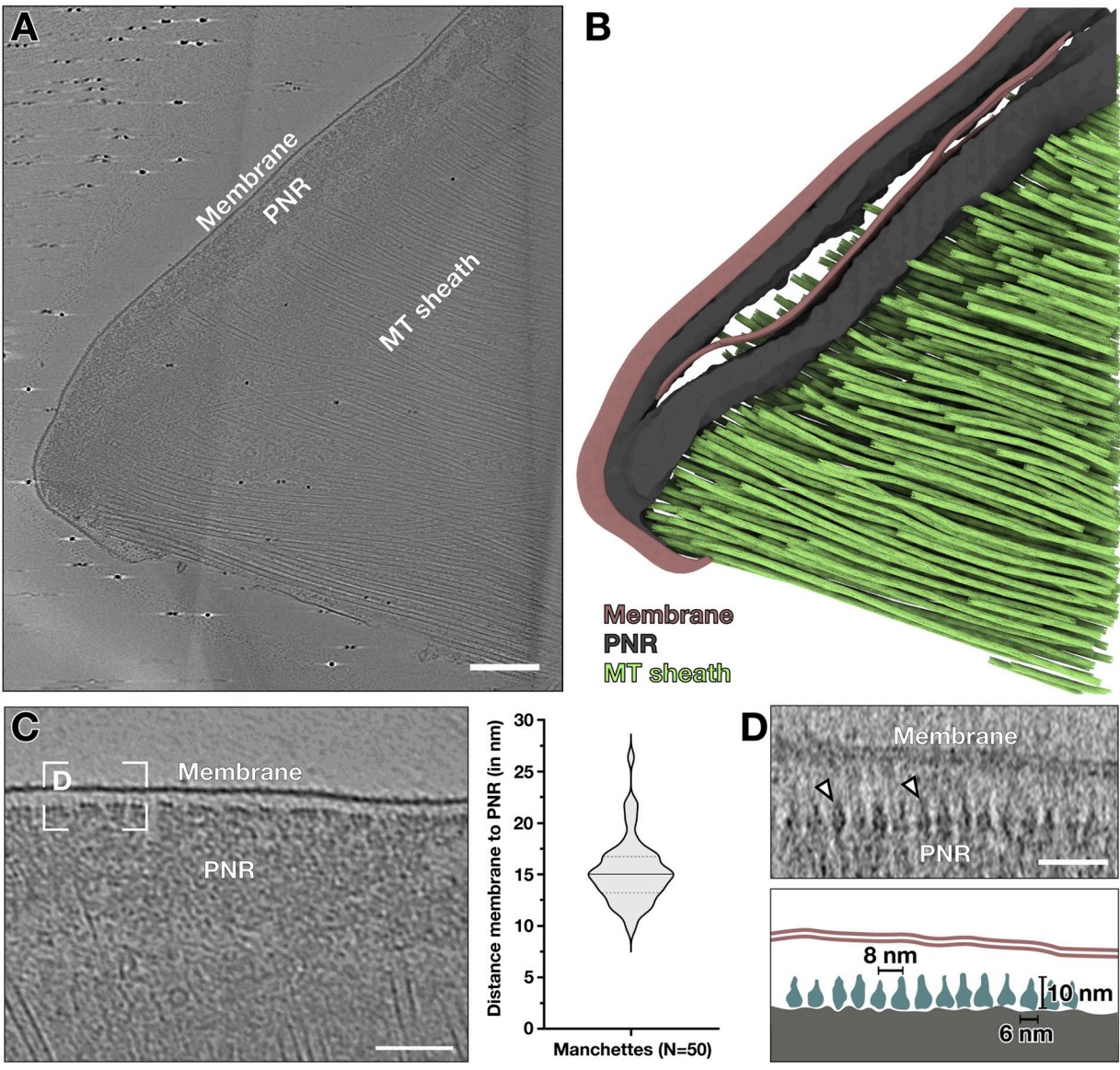

**Figure 3. A membrane is laminated to the perinuclear ring.**
**(A)** A tomographic slice resolving a membrane that envelopes the dense perinuclear ring (PNR) and vesicles that are associated with the microtubules. Scale bar, 250 nm. **(B)** Segmentation of the tomogram in (A). Membrane (brown), perinuclear ring (dark grey), and the MT sheath (green). **(C)** Distance measurements between the inner membrane leaflet and the top border of the perinuclear ring. **(D)** Tomographic slice revealing regular densities repeating every 8 nm under the membrane (white arrows). Scale bar, 25 nm. Annotation colour scheme: PNR, grey; membrane, brown; regular densities, teal.

adaptors BICD2, JIP3, and HOOK3 were, however, absent, further suggesting specializations in the dynactin components of spermatids (Table S2). Nevertheless, further investigations are required to determine whether spermatids possess specialized components of the dynactin complex.

In the manchette, the MT plus ends are organized unidirectionally, with the plus ends at the perinuclear ring and the minus ends leading towards the developing axoneme (Akhmanova et al, 2005; Lehti & Sironen, 2016). We determined MT directionality via protofilament skew (Fig S4) and showed that the putative dynactin densities were always oriented towards the plus end of the MTs, whereas the dynein MT domains

were oriented towards the MT minus ends (Fig 5). This is in agreement with the minus-end–directed movement of cytoplasmic dynein-1 (Burgess et al, 2003; Canty & Yildiz, 2020; Canty et al, 2021). The MT unidirectionality presents a problem for the efficient coordination of macromolecular IMT. In cilia, microtubular unidirectionality is circumvented by the formation of intra-flagellar transport (IFT) trains that combine dyneins, kinesins, and IFT complexes A and B (Lacey et al, 2023). In the manchette, subunits of IFT complexes A and B were amongst the first proteins hypothesized to play a role in IMT, and parallels between IMT and IFT have been suggested (Kierszenbaum, 2002). However, in the proteomics data of isolated rat manchettes, IFT

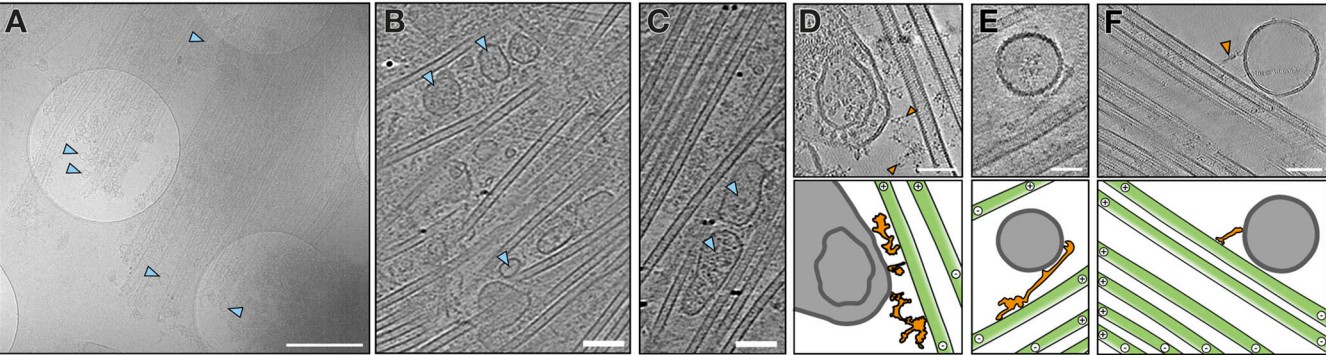

**Figure 4. Vesicles are associated with the manchette.**
**(A)** A low-magnification projection image of the manchette periphery revealing many vesicles colocalizing with MTs. Scale bar, 1 $\mu$m. **(B, C)** Tomographic slices showing many vesicles close to the MTs (blue triangles). Scale bars, 100 nm. **(D, E, F)** Tomographic slices (top) and annotation (bottom) showing vesicles (grey) connected to a microtubule (green) through a slender density (orange). Scale bars, 50 nm.

subunits were about 10–100 times less abundant than microtubule motors dynein-1 and kinesin (Fig 5F, blue circles), including the previously reported IFT subunits IFT88 and IFT20 (Yap et al, 2022). Furthermore, we did not observe densities resembling IFT trains like in cilia (Lacey et al, 2023) in our tomograms. Thus, the concrete role of IFT complex subunits in IMT remains unclear. Instead, MT unidirectionality in the manchette must be navigated differently compared with the axoneme, likely involving plus-end–directed kinesins.

### Actin filaments are part of the manchette

In tomograms of isolated manchettes, we occasionally observed non-MT filaments (Fig 6). 3D reconstruction of these filaments revealed dimensions and helical twist of F-actin (Figs 6A and B and S3B). In addition, in our proteomics data, actin isoforms alpha-actin-2 and beta- and gamma-actin were highly abundant, supporting the assignment and abundance of F-actin (Fig 5F, Table S1). The actin filaments were often clustered parallel to the MTs (Fig 6A and D). These clusters comprised 2 to 10 single actin filaments densely packed on average about 11 nm apart (Fig 6A and C). Actin filament bundle formation requires crosslinking proteins (Rajan et al, 2023). Although we have not observed densities between the actin filaments in the manchette, our proteomics analysis detected the actin crosslinking proteins espin, $\alpha$-actinin, and fascin (Table S1). Both $\alpha$-actinin and fascin showed a relatively low abundance of 100–1,000 times less than actin, but espin was only four times lower in abundance compared with actin. The relatively constant distance between the actin filament clusters in the manchette is very similar to the distance reported for espin- and fascin-bundled actin filaments (10–11 nm) (Kitajiri et al, 2010; Jansen et al, 2011; Rajan et al, 2023).

Aside from the actin filament clusters, we observed single actin filaments in between the manchette MTs (Fig 6E–K). In agreement with a potential role of actin in IMT, myosins were only slightly lower in abundance than kinesins and dynein subunits in the proteomics data (Fig 5F, Table S2); yet, we were unable to confirm myosin binding to actin filaments from our tomograms. We only

found a density bound to an actin filament in one instance (Fig 6E). Thus, it was impossible to assign the identity of the density to myosins. The single actin filaments did not always run parallel to the MTs. Instead, they, in some instances, seemed to interact with MTs seemingly with a protein density binding on one or more protofilaments of the MT (Fig 6I–K), hinting towards a possible direct interaction between actin and MTs.

## Discussion

The manchette is a unique MT-based structure of critical importance for shaping sperm heads (Rattner & Brinkley, 1972; Russell et al, 1991; O'Donnell & O'Bryan, 2014; Dunleavy et al, 2019a). The scaffold comprises an amorphous protein matrix, the perinuclear ring, and thousands of parallel MTs that emanate from the perinuclear ring. Our data showed a correlation between the thickness of the perinuclear ring and constriction/elongation of the manchette. Thickening of the perinuclear ring may be essential for nuclear shaping during spermiogenesis. The isolated manchettes retained a membrane that lined the perinuclear ring. Plastic-embedded EM imaging of spermatids suggests that this membrane might be the plasma membrane of the spermatid bulging over the perinuclear ring (Rattner & Brinkley, 1972; O'Donnell, 2014; Dunleavy et al, 2019a). We propose that the membrane is maintained because of a regular protein array we observed on top of the perinuclear ring that might tether the perinuclear ring to the membrane. The exact molecular composition of the perinuclear ring is unknown, but previous studies have suggested keratin 9 and $\delta$-tubulin as its components (Mochida et al, 1998; Kato et al, 2004). Recent proteomics data of isolated manchettes did not find the presence of either of these proteins (Hu et al, 2023). Instead, one of the most abundant proteins detected was KIF27. Interestingly, KIF27 colocalizes with the perinuclear ring (Nozawa et al, 2014) and has been shown to suppress MT growth in vitro (Yue et al, 2018), highlighting its role as a MT regulator. However, its function in the manchette and perinuclear ring is unclear and should be further studied in the future.

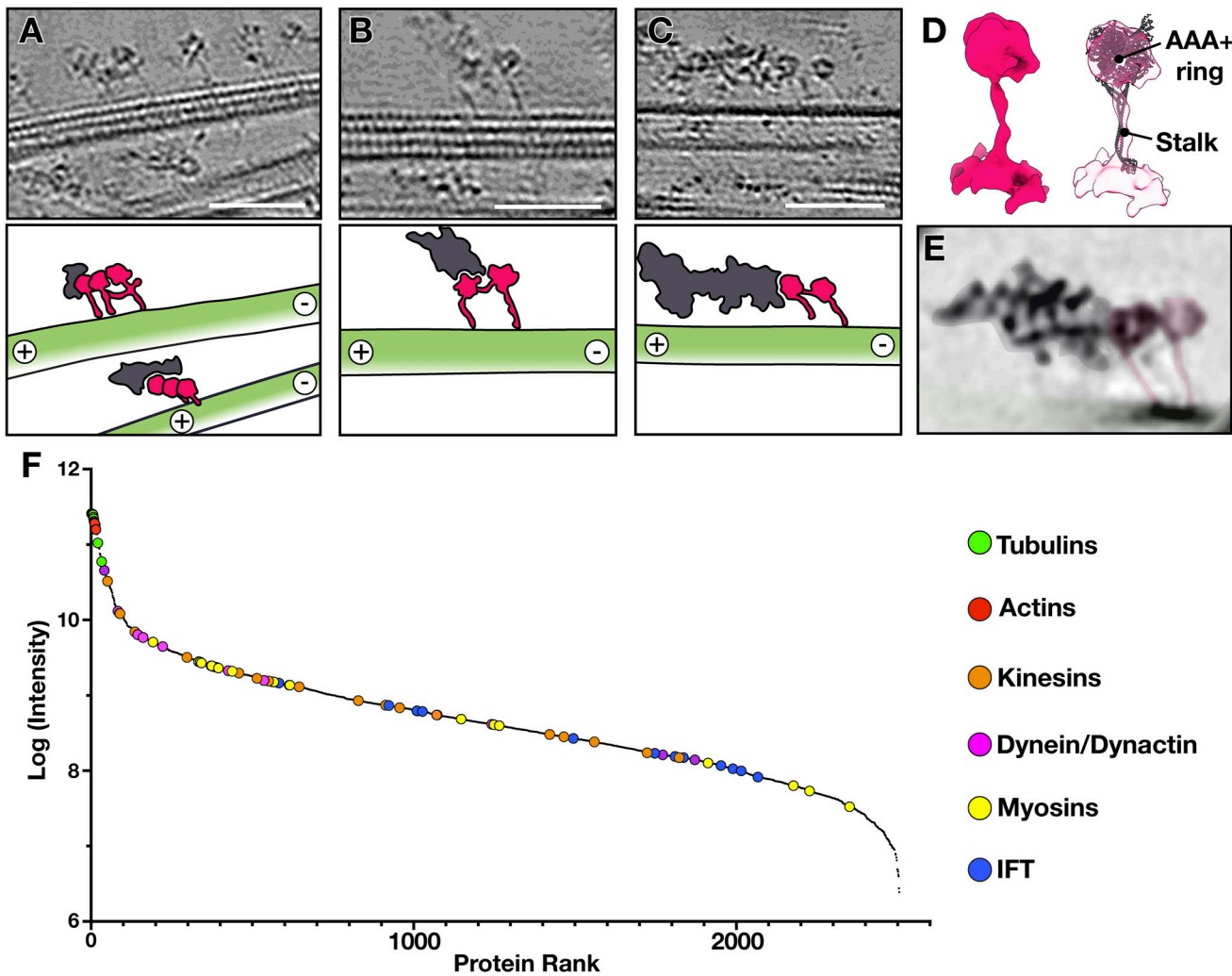

**Figure 5. Dynein on manchette MTs.**
**(A, B, C)** Tomographic slices (top) and annotations (bottom) revealing densities associated with the MTs (green). Scale bars, 50 nm. **(D)** 3D reconstruction of the densities found in (A, B, C) with cytoplasmic dynein I motor domain (PDB: 5NVU) fitted into the map (right). **(E)** A slice through a reconstruction of the cytosolic dynein–dynactin complex (EMD-7000) with dynein motor domains coloured pink, and the dynein tails and dynactin complexes are coloured dark grey. **(F)** Relative abundance of tubulin, actin, and components of the transport machinery in isolated manchettes. Tubulins (green) and actins (red) show the highest abundance. Components of the microtubule motor proteins kinesins (orange) and dynein (pink) are similar in abundance. Components of the actin motor proteins myosins (yellow) are less abundant than dyneins and kinesins. IFT subunits (blue) show the lowest abundance.

The manchette has been proposed to play a crucial role as a platform for bidirectional transport, ensuring the timely delivery of macromolecules and vesicles in the spermatid requiring for the development of sperm cells (Kierszenbaum, 2002). Here, we showed the association of vesicles with the manchette in intact spermatids and isolated manchettes. The 3D visualization enabled to observe dynein bound to manchette MTs, consistent with a proposed active role of dynein in IMT (Kierszenbaum et al, 2011a), and proteomics revealed the presence of nearly all dynactin components and several cargo linkers. However, we cannot exclude that certain transport components are lost upon isolation of the manchettes, and whether missing dynactin components DCTN4 and ARP11 are indeed absent during spermiogenesis needs to be tested in intact spermatids.

Our data did not allow us to corroborate a clear role of IFT subunits in IMT, as we did not observe densities that resemble ciliary IFT trains. Thus, the specific role of IFT complex subunits in IMT remains unclear; instead, MT unidirectionality in the manchette appears to require the existence of a different molecular mechanism. It was shown that IFT-88 plays a role outside IFT trains in relocating essential factors of cytokinesis during cell division (Taulet et al, 2017). IFT-88 might, in a similar way, relocate essential factors for spermiogenesis in the manchette, but this hypothesis awaits experimental evidence. Alternatively, IFT components might play a role in the manchette in a way that is not yet understood. Our data support a mechanism in which dynein–dynactin complexes transport macromolecules downwards to the developing sperm tail, whereas kinesins separately transport macromolecules upwards to the spermatid nucleus. One limitation of our study is

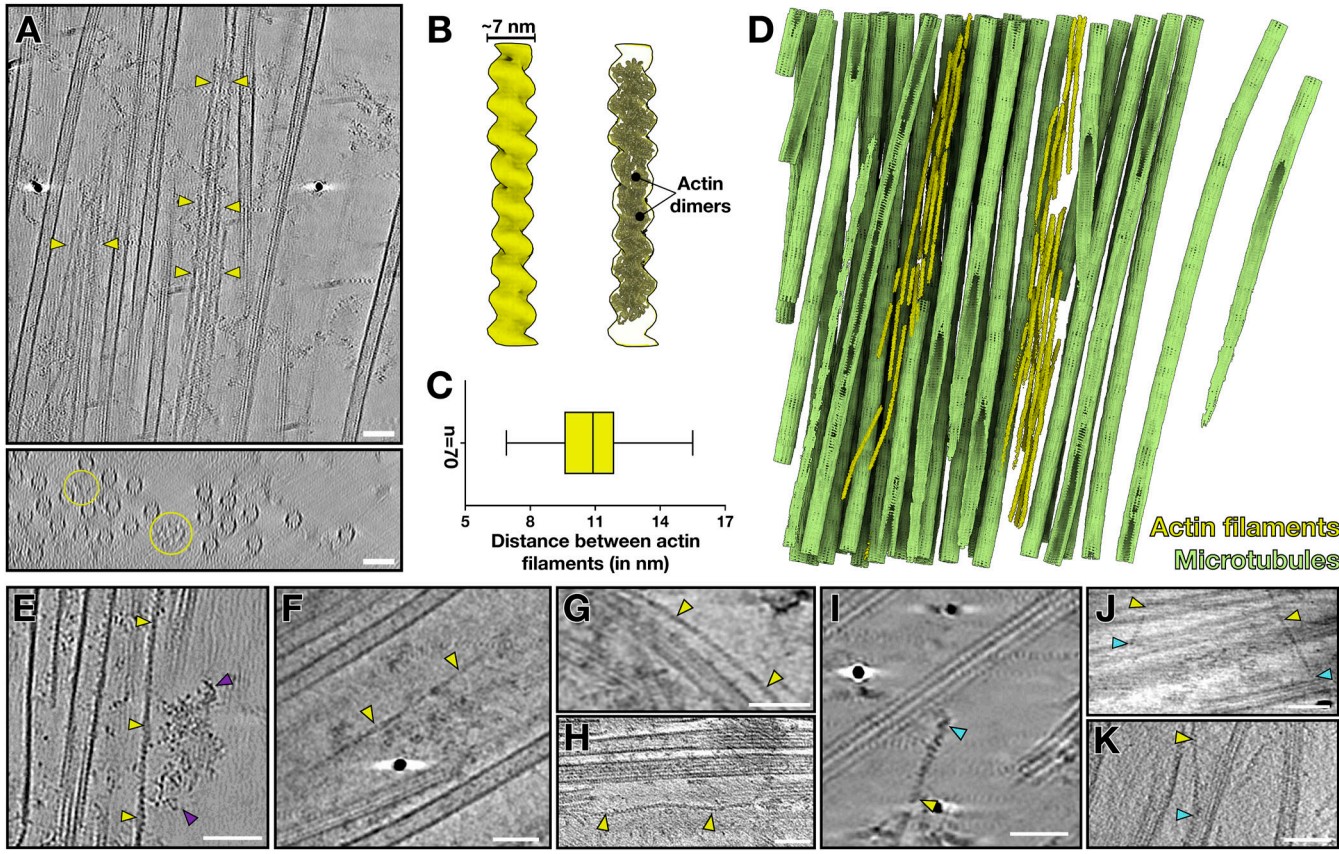

**Figure 6. Actin filaments are intercalated in the manchette.**
**(A)** Orthogonal slices from a tomogram showing non-MT filaments (yellow arrowheads and yellow circles). Scale bar, 50 nm. **(B)** 3D reconstruction and fitting of an F-actin structure (PDB: 7BT7). **(C)** Quantification of actin-to-actin filament distances. **(D)** Segmentation of the tomogram in (A) with actin filaments (yellow) running parallel and in between MTs (green). **(E)** Tomographic slice of a single actin filament (yellow arrowheads) bound by a large density (violet arrowheads). Scale bar, 50 nm. **(F, G, H)** Tomographic slices showing actin single filaments (yellow arrowheads) running parallel to MTs. Scale bars, 50 nm. **(I, J, K)** Tomographic slices with actin singlet filaments interacting with MTs. The interactions are marked in cyan. Scale bar in (I), 25 nm; scale bars in (J, K), 50 nm.

that connections between the manchette and the nuclear envelope are lost upon isolation, and we therefore cannot provide conclusions on the role of dynein–dynactin complexes in nuclear shaping. It is, however, likely that in spermatids, dynein motors bound to the nuclear envelope could exert a downward force that could facilitate nuclear shaping. In this regard, our proteomics data revealed the presence of dynactin–cargo linkers NUMA1 and RAB11FIP1, which are unstudied in the context of the manchette. Interestingly, NUMA1 was recently shown to activate dynein and connect it to microtubule minus ends (Colombo et al, 2025). Dynein–dynactin–NUMA1 complexes might have a possible role in manchette organization by bundling the MT minus ends at the caudal end of the manchette.

We found that actin filaments remain associated with the manchettes after isolation and are present as clusters of parallel filaments and single filaments running between MTs. Actin bundles in other cells are often associated with mechanical functions, for example, generating forces relevant to cellular processes like cell migration, adhesion, and cell division (De La Cruz & Gardel, 2015; Rückerl et al, 2017; Rajan et al, 2023). The role of actin bundles in the manchette is, however, not fully understood. Although our study unequivocally confirms the presence of actin filaments, we cannot conclude on their function in the manchette. Actin filament bundles are formed by cross-linking proteins (Jansen et al, 2011; Yang et al, 2022; Rajan et al, 2023), and our proteomics data show the crosslinking protein espin in high relative abundance. Actin filament distances agree with those found in espin–actin and fascin–actin bundles (Kitajiri et al, 2010; Rajan et al, 2023). Depleting these actin-bundling proteins in spermatids with RNAi might help clarify the importance of actin bundles during spermiogenesis. Actin has previously been proposed to facilitate accurate short-distance IMT (Kierszenbaum, 2002). The here-described actin singlet filaments could hypothetically function in cargo transport. We did not observe an apparent decoration allowing unequivocal identification of actin-based transport in the manchette. In some instances, we observed interactions between an actin single filament and MTs that could potentially facilitate the proposed "track" changes between MTs and actin. In this regard, mouse knockout experiments of actin–microtubule crosslinking proteins, such as MACF1, could unveil the role of actin-to-microtubule interaction in the manchette.

To summarize, we revealed the architecture of the manchette and its perinuclear ring and showed the association of vesicles with the manchette in situ and in isolated manchettes. In isolated manchettes, we also found densities that directly link vesicles and MTs. We further revealed dynein–dynactin complexes binding to manchette MTs and suggest the possibility of specializations in dynactin complex components and cargo linkers in spermatids. We further find F-actin in single filaments and filament clusters intercalating with the MTs, hinting towards two separate roles of actin filaments in the manchette. Thus, taken together, our results provide new insights into the manchette's architecture and function as a transport scaffold, highlighting its significance for the shaping of mammalian sperm cells during spermiogenesis.

# Materials and Methods

### Mouse spermatid purification and preparation for FIB milling

Testes were dissected from freshly asphyxiated mice. After removing the tunica albuginea, the seminiferous tubules were disassociated and mechanically cut into small pieces with a sharp razor blade. The tissue of each testis was pelleted at 220$g$ for 10 min, the supernatant was carefully discarded, and the pellet was resuspended in a mixture of 8 ml of collagenase (0.5 mg/ml, #9001-12-1; Sigma-Aldrich) and DNase I (0.5 mg/ml, #9003-98-9; Sigma-Aldrich). Enzymatic digestion was performed at 37°C for 20 min to remove connective tissue and interstitial cells. The suspension was then centrifuged for 10 min at 220$g$ at RT. The pellet was thoroughly resuspended in 10 ml DMEM and repeatedly pipetted to dissociate the cells mechanically. The cell suspension was filtered through a 30-$\mu$m nylon sieve to remove remaining clumps of cells and undigested tissue. The filtered cell suspension was loaded onto a modified STAPUT gradient of 100 ml 2–4% wt/vol bovine serum albumin/DMEM as described previously (Dunleavy et al, 2019b; Cafe et al, 2020). Germ cell stages were fractionated based on size and density via velocity sedimentation for 3.5 h at RT. Seven fractions were collected: 10 ml for fractions 1–3 and 5–7 and 15 ml for fraction 4. Cells from fractions 2 (spermatocytes), 5 (round spermatids), 6 (round and elongating spermatids), and 7 (elongating spermatids and spermatozoa) were harvested at 220$g$ for 10 min at RT and washed once with DMEM. Spermatids were resuspended in DMEM supplemented with 10% dextran. Directly thereafter, 4 $\mu$l of cells was applied to glow-discharged Quantifoil R 2/1 200-mesh copper holey carbon grids at a concentration of $2 \times 10^7$ cells/ml. The grids were blotted from the back for 4–5 s with Whatman 1 filter paper on a manual plunger (MPI Martinsried). Grids were plunged into liquid ethane cooled to liquid nitrogen temperatures. After plunge-freezing, grids were stored under liquid $N_2$.

### Cryo-FIB milling

Grids were loaded into an Aquilos dual-beam cryo-focused ion beam/scanning electron microscope (cryo-FIB/SEM) (Thermo Fisher Scientific) operated at −175°C. SEM imaging was performed at 2 kV and 13 pA, whereas FIB imaging for targeting was performed at 30 kV and 10 pA. After loading, grids were coated with an organometallic platinum layer to reduce charging and protect the milling front. FIB milling was typically performed with a stage tilt of 18°, and thus, lamellae were inclined at 11° relative to the grid. Lamellae were milled with expansion segments (Wolff et al, 2019). Each lamella was milled in four stages: an initial rough mill at 500 pA beam current, an intermediate mill at 300 pA, a fine mill at 100 pA, and a polishing step at 30 pA. After polishing, lamellae were sputter-coated with a thin layer of metallic platinum to reduce charging during cryo-ET imaging.

### Cryo-ET data collection of FIB-milled spermatids

Tilt series were acquired on a 200 kV Talos Arctica (Thermo Fisher Scientific) equipped with a post-column energy filter (Gatan) operated in zero-loss imaging mode with a 20 eV energy selecting slit. All images were recorded on a K2 Summit direct electron detector (Gatan) in counting mode with a pixel size of 4.473 Å. Tilt series were collected using SerialEM (Mastronarde, 2003) at a target defocus of between −3 and −5 $\mu$m. Tilt series were typically recorded using strict dose-symmetric schemes, spanning ±51° in 3° increments, with the total dose limited to ~100 e⁻/Å².

### Rat manchette isolation

Isolation of manchettes from rat testes was based on a previously published protocol (Mochida et al, 1998). Briefly, testes were prepared from the abdomen of freshly asphyxiated rats (Wistar, Crl:CD(SD), RJHan:WI, Lister Hooded) in eight biological replicates. Fat tissue and epididymis were carefully removed using sharp scissors. The testes were then transferred to a culture dish containing PBS on ice, and the tunica albuginea was removed using tweezers. The seminiferous tubules were briefly washed and then transferred into 10 ml of microtubule-stabilizing buffer (MSB: 25 mM Hepes, 2.5 mM MgSO$_4$, 2.5 mM EGTA, 15 mM KCl, 5 mM DTT, 0.1 mM GTP, 20 $\mu$M Taxol, 1% Triton X-100, 1 tablet of cOmplete mini-protease inhibitor [Roche]). Next, the seminiferous tubules were chopped using a razor blade and then repeatedly pipetted with a 1-ml pipette to break up the tissue and separate the manchettes. The suspension was filtered through 100-, 30-, and 10-$\mu$m-mesh-sized cell strainers and collected in a 50-ml Falcon tube. The cell suspension was further filtered through 7.5 g of 212- to 300-$\mu$m glass beads packed on a 20-ml syringe and prewashed with ice-cold PBS. After filtration, 6 ml of the suspension was added to 44 ml of cold 2.5 M sucrose. The resulting 50 ml suspension was split equally into two open-top thin-wall ultracentrifugation tubes (#344058; Beckmann Coulter). The mixture was layered with 6 ml of cold 2.05 M sucrose followed by 6 ml of cold 1 M sucrose and centrifuged at 85,000$g$ for 110 min at 4°C using a SW32 rotor in a Beckman Coulter Optima XPN-80 ultracentrifuge. Approximately 3 ml of intact manchettes was harvested from each tube at the interface between the 1 and 2.05 M sucrose and transferred into 12 ml cold PBS. The mixture was then centrifuged at 1,000$g$ for 30 min at 4°C, the supernatant was discarded, and the pellet was resuspended in 500 $\mu$l PBS. The manchette-containing pellets were washed three

times by resuspension in cold PBS and directly used for cryo-ET and proteomics sample preparation.

### Cryo-ET sample preparation of manchettes and data collection

Isolated manchettes were mixed with BSA-conjugated gold beads (Aurion) in a 4:1 ratio, and 3.5–4 $\mu$l of the mixture was applied to glow-discharged Quantifoil R 2/1 200-mesh holey carbon grids. The grids were blotted from the back for 3–4 s with Whatman 1 filter paper on a manual plunger (MPI Martinsried). Grids were plunged into liquid ethane cooled to liquid nitrogen temperatures. Cryo-ET data of manchettes were collected on a 200 kV Talos Arctica (Thermo Fisher Scientific) with a pixel size of 2.17 Å or on a 300 kV Titan Krios with a pixel size of 6.32 Å. A total of 125 tilt series were collected from a total of four grids from three separate manchette preparations. Tilt series were collected using SerialEM (Mastronarde, 2003) at a target defocus of between −2 and −4 $\mu$m. Tilt series were typically recorded using strict dose-symmetric schemes, spanning ± 51° in 3° increments, with the total dose limited to ~100 e$^-$/Å$^2$.

### Tomogram reconstruction

Motion between individual frames was corrected using Motion-Cor2 1.2.1 (Zheng et al, 2017). Tomogram reconstruction was performed in either IMOD or AreTomo (Mastronarde & Held, 2017; Zheng et al, 2022). In IMOD, four-times binned tomograms were reconstructed using weighted back-projection, with a SIRT-like filter applied for visualization/segmentation and CTF-corrected using IMODs ctfphaseflip function for subtomogram averaging in PEET 1.13.0 (Nicastro et al, 2006; Heumann et al, 2011). In AreTomo, four-times binned tomograms were reconstructed using weighted back-projection. AreTomo reconstructed tomograms were denoised with CryoCare (Buchholz et al, 2019) for visualization purposes.

### Determination of MT directionality

Individual MTs were manually traced in IMOD, and model points were imported to the software package cylindra (Liu et al, 2024 Preprint). In cylindra, the protofilament skew and numbers were determined for each MT using local averages. A clockwise protofilament skew is a view from the minus-end to plus-end direction. A counterclockwise skew is a view from the plus-end to minus-end direction.

### Subtomogram averaging of dynein motor domains

Individual particles were manually picked in IMOD using low-pass–filtered subtilt tomograms four times and binned to a final pixel size of 8.68 Å. Two points indicating the top and bottom of the dynein motor domain relative to the microtubule were picked per particle. 49 particles were picked from four tomograms and used for subtomogram averaging in PEET 1.13.0 (Nicastro et al, 2006; Heumann et al, 2011). Initial particle orientation and rotation axes of particles were generated relative to the MT via the SpikeInit function of PEET and used as initial information for the alignment.

Resolution was estimated using the Fourier shell correlation at a cut-off of 0.5.

### Length and distance measurements and statistical analysis

All measurements were taken in IMOD as the distance between two points in the tomograms. For membrane-to-PNR distances, 50 manchettes were assessed. The distance at five locations was measured within each manchette, and the mean was used as the value per manchette. PNR thickness was measured at five locations, and the mean value was used for each PNR to yield a total of 75 data points. Statistical analysis was carried out in GraphPad. The mean and standard deviations were determined for all measurements and are given in the figures.

### Subtomogram averaging of actin filaments

Individual actin filaments were manually traced in IMOD, and model points were added every 3.6 nm using addModPts. Subtomogram averaging with missing wedge compensation was performed using PEET 1.13.0. Alignments were generally performed first on four-times binned data. Subtomograms of ~16 nm × 16 nm × 16 nm were computationally aligned and averaged in steps of tighter angular and translational search ranges. A tight cylindrical mask was used during alignment to exclude signal from neighbouring actin filaments. 928 particles from three tomograms contributed to the final average with a resolution of 30 Å. Resolution was estimated using the Fourier shell correlation at a cut-off of 0.5.

### Distance measurement between actin filaments

Tomogram slices perpendicular to actin bundles were generated using the IMOD slicer. Distances between actin filaments were measured across 70 positions as distance between two points in the centre of the actin filament. Measurements were carried out across two tomograms and three total bundles directly using the IMOD measurement tool. Distances were plotted using GraphPad Prism 10.

### Proteomics sample preparation

Single-pot solid-phase–enhanced sample preparation (SP3) (Hughes et al, 2019) was used for sample processing for proteomics analysis. Samples were lysed with 20 $\mu$l of lysis buffer (1% [wt/vol] sodium dodecyl sulphate, 100 mM Hepes, pH 8, 1% [vol/vol] EDTA-free complete protease inhibitor [Roche]) and sonicated in the Bioruptor (Diagenode) for 15 cycles of 15-s on/off intervals. Benzonase (purity > 90%; Millipore) was added to 1% (vol/vol) to remove RNA or DNA contaminants, and the sample was incubated at 37°C for 15 min. The lysates were centrifuged for 10 min at 18,000$g$ at 4°C, and the supernatant was collected. A bicinchoninic acid assay was performed to estimate protein concentration. Each sample was split into three equal parts and diluted with lysis buffer.

Samples were reduced and alkylated for 5 min at 95°C by adding tris(2-carboxyethyl)phosphine and chloroacetamide to

a final concentration of 40 and 10 mM, respectively. Prewashed Sera-Mag paramagnetic beads (Thermo Fisher Scientific) were added at a 1:10 bead-to-protein ratio, and protein binding was induced by adding ethanol to 75% (vol/vol). Samples were incubated for 20 min at 1,000 rpm at RT in a thermo-shaker. After incubation, the beads were rinsed in the magnetic rack twice with 200 $\mu$l of 80% (vol/vol) ethanol and once with 200 $\mu$l of 100% (vol/vol) acetonitrile. The proteins were resuspended in 100 mM ammonium bicarbonate, and samples were sonicated in a water bath for 2 min. Protein digestion was achieved by adding trypsin and LysC at 1:25 and 1:75 enzyme-to-protein ratios, respectively. Samples were incubated at 37°C overnight (ca. 16 h) at 1,000 rpm in a thermo-shaker.

Samples were centrifuged for 2 min at 18,000$g$ and acidified with 5% (vol/vol) trifluoroacetic acid. The beads were immobilized in the magnetic rack, and the peptide solution was recovered. Finally, sample clean-up was performed using Oasis HLB 96-well $\mu$Elution Plate (Waters). Peptides were dried in a SpeedVac concentrator and resuspended in 0.1% formic acid before liquid chromatography with tandem mass spectrometry ((LC)-MS/MS) analysis.

### LC-MS/MS analysis

Peptide samples were analysed using a nanoUltimate 3000 UHPLC (Thermo Fisher Scientific) coupled to an Orbitrap Exploris 480 mass spectrometer (Thermo Fisher Scientific). The sample (1 $\mu$g) was injected at 3 $\mu$l/min for 1 min into a trap column (Acclaim Pepmap 100 C18, 5 mm × 0.3 mm, 5 $\mu$m; Thermo Fisher Scientific). The peptide separation was performed in a 50-cm-long analytical column with a 75 $\mu$m inner diameter packed in-house with C18 beads (C18, 1.9 $\mu$m; Reprosil) at a column temperature of 32°C. Peptides were eluted at 300 nl/min with a total run time of 90 min. Gradient separation on the analytical column was as follows: 9% B (80% acetonitrile and 0.1% formic acid) for 1 min, from 9% to 13% B in 1 min, from 13% to 44% in 65 min, from 44% to 55% in 5 min, from 55% to 99% in 3 min, 99% B for 5 min, and 9% B for 10 min.

The MS acquisition method was a data-dependent acquisition mode using the Orbitrap analyser at 60 K mass resolution in the scan range 375–1,600 m/z, with automatic maximum injection time and a standard AGC target. Ions subjected to MS2 were filtered based on dynamic exclusion with a mass tolerance of 10 ppm. Peptides were fragmented with HCD of 28% and mass resolution of 15 K.

### MS data analysis

The raw files were analysed with FragPipe version 20.0 (Kong et al, 2017). Data were searched against the reviewed and unreviewed UniProt rat proteome (UniProt Proteome ID: UP000002494, downloaded November 2023), and known contaminants. The precursor and the fragment mass tolerance were set to 20 ppm. Full tryptic digestion with a maximum of three missed cleavages was allowed. Carbamidomethylation (Cys) was set as a fixed modification, and oxidation (Met) and acetylation (protein N terminus) were set as a dynamic modification. The false discovery rate (FDR) was set to 1%. Further data analysis was performed in the RStudio programming language environment. Protein intensities were extracted from FragPipe output. The data were filtered, removing proteins without quantitative values, with only shared peptides, and contaminants. Only proteins quantified in two out of three replicates were further used for the analysis.

## Data Availability

All LC-MS/MS data have been deposited to the ProteomeXchange Consortium via the PRIDE partner repository with the dataset identifier PXD055902.

## Supplementary Information

## Acknowledgements

We thank the Instantie voor Dierenwelzijn (IvD) Utrecht and Ate Bijlsma for providing and euthanizing the rats. Cryo-ET data were collected at the Utrecht University Electron Microscopy Centre and at the Netherlands Centre for Electron Nanoscopy (NeCEN). We thank Dr. M Vanevic for computational support at Utrecht University, and acknowledge Dr. SC Howes, M Bergmeijer, Ingr. C Schneijdenberg, and J Meeldijk for management and maintenance of the Utrecht University EM Centre. We thank Willem Noteborn and NeCEN for help with data collection. This work benefited from the Netherlands Electron Microscopy Infrastructure (NEMI), project number 184.034.014 of the National Roadmap for Large-Scale Research Infrastructure of the Dutch Research Council (NWO). The proteomics analysis was supported by the Dutch Research Council (NWO) through funding for the Netherlands Proteomics Centre through the X-omics Road Map program (project 184.034.019). T Zeev-Ben-Mordehai was funded by the European Research Council (ERC-2022-COG project 101088673).

### Author Contributions

JH Judernatz: conceptualization, formal analysis, visualization, methodology, and writing—original draft, review, and editing.
L Pérez Pañeda: formal analysis, methodology, and writing—review and editing.
T Kadavá: formal analysis, methodology, and writing—review and editing.
AJR Heck: supervision, funding acquisition, and writing—review and editing.
T Zeev-Ben-Mordehai: conceptualization, formal analysis, supervision, funding acquisition, validation, visualization, methodology, project administration, and writing—original draft, review, and editing.

### Conflict of Interest Statement

The authors declare that they have no conflict of interest.

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
