## [Reviewer comments · Life Science Alliance]

Life Science Alliance

Characterisation of the Spermatid Manchette Architecture and its Role as Transport Scaffold

Jo Judernatz, Laura Perez Paneda, Tereza Kadavá, Albert Heck, and Tzviya Zeev-Ben-Mordehai

DOI: <https://doi.org/10.26508/lsa.202503415>

Corresponding author(s): Tzviya Zeev-Ben-Mordehai, Utrecht University

Review Timeline:

Submission Date:	2025-06-10
Editorial Decision:	2025-07-08
Revision Received:	2025-07-16
Accepted:	2025-07-21

Scientific Editor: Tim Fessenden

Transaction Report:

Please note that the manuscript was reviewed at *Review Commons* and these reports were taken into account in the decision-making process at *Life Science Alliance*.

Reviews

Review #1

1. Evidence, reproducibility and clarity:

Evidence, reproducibility and clarity (Required)

In this manuscript the authors have done cryo-electron tomography of the manchette, a microtubule-based structure important for proper sperm head formation during spermatogenesis. They also did mass-spectrometry of the isolated structures. Vesicles, actin and their linkers to microtubules within the structure are shown.

****Major:****

The data the conclusions are based on seem very limited and sometimes overinterpreted. For example, only one connection between actin and microtubules was observed, and this is thought to be MACF1 simply based on its presence in the MS.

Another, and larger concern, is that the authors do a structural study on something that has been purified out of the cell, a process which is extremely disruptive. Vesicles, actin and other cellular components could easily be trapped in this cytoskeletal sieve during the purification process and as such, not be bona fide manchette components. This could create both misleading proteomics and imaging. Therefore, an approach not requiring extraction such as high-pressure freezing, sectioning and room-temperature electron tomography and/or immunoEM on sections to set aside this concern is strongly recommended. As an additional bonus, it would show if the vesicles containing ATP synthase are deformed mitochondria.

****Minor:****

Line 99: "to study IMT with cryo-ET, manchettes were isolated ...(insert from which organism)..."

Line 102 "...demonstrating that they can be used to study IMT".. can the authors please clarify?

Line 111 "densities face towards the MT plus-end" How can a density "face" anywhere? For this, it needs to have a defined front and back.

Line 137: is the "perinuclear ring" the same as the manchette?

Figure 2B: How did the authors decide to not model the electron density found between the vesicle and the MT at 3 O'clock? Is there no other proteins with a similar lollipop structure as ATP synthase, so that this can be said to be this protein with such certainty?

Line 189: "F-actin formed organized bundles running parallel to mMTs" - this observation needs confirming in a less disrupted sample.

Line 242 remove first comma sign

Line 363 "a total of 2 datasets" - is this manuscript based on only two tilt-series? Or two datasets from each of the 4 grids? In any case, this is very limited data.

2. Significance:

Significance (Required)

The article is very interesting, and if presented together with the suggested controls, would be informative to both microtubule/motorprotein researchers as well as those trying studying spermatogenesis.

3. How much time do you estimate the authors will need to complete the suggested revisions:

Estimated time to Complete Revisions (Required)

(Decision Recommendation)

Between 1 and 3 months

4. Review Commons values the work of reviewers and encourages them to get credit for their work. Select 'Yes' below to register your reviewing activity at Web of Science Reviewer Recognition Service (formerly Publons); note that the content of your review will not be visible on Web of Science.

Yes

Review #2

1. Evidence, reproducibility and clarity:

Evidence, reproducibility and clarity (Required)

The manchette appears as a shield-like structure surrounding the flagellar basal body upon spermiogenesis. It consists of a number of microtubules like a comb, but actin (Mochida et al. 1998 Dev. Biol. 200, 46) and myosin (Hayasaka et al. 2008 Asian J. Androl. 10, 561) were found, suggesting transportation inside the manchette. Detailed structural information and functional insight into the manchette was still awaited. There is a hypothesis called IMT (intra manchette transport) based on the fact that manchette and IFT (intraflagellar transport) share common components (or homologues) and on their transition along the stages of spermiogenesis. While IMT is considered as a potential hypothesis to explain delivery of centrosomal and flagellar components, no one has witnessed IMT at the same level as IFT. IMT has never been purified, visualized in motion or at high resolution.

This study for the first time visualized manchette using high-end cryo-electron tomography of isolated manchettes, addressing structural characterization of IMT. The authors successfully microtubular bundles, vesicles located between microtubules and a linker-like structure connecting the vesicle and the microtubule. On multilamellar membranes in the vesicles they found particles and assigned them to ATPase complexes, based on intermediate (~60Å) resolution structure. They further identified interesting structures, such as (1) particles on microtubules, which resemble dynein and (2) filaments which shows symmetry of F-actin. All the molecular assignments are consistent with their proteomics of manchettes.

Their assignment of ATPase will be strengthened by MS data, if it proves absence of other possible proteins forming such a membrane protein complex.

They discussed possible role of various motor proteins based on their abundance (Line 134-151, Line 200). This makes sense only with a control. Absolute abundance of proteins would not necessarily present their local importance or roles. This reviewer would suggest quantitative proteomics of other organelles, or whole cells, or other fractions obtained during manchette isolation, to demonstrate unique abundance of KIF27 and other proteins of their interest. A single image from a tomogram, Fig.6B, is not enough to prove actin-MT interaction. A gallery and a number (how many such junctions were found from how many MTs) will be necessary.

****Minor points:****

Their manchette purification is based on Mochida et al., which showed (their Fig.2) similarity to the in vivo structure (for example, Fig.1 of Kierszenbaum 2001 Mol. Reproduc. Dev. 59, 347). Nevertheless, since this is not a very common prep, it is helpful to show the wide view (low mag cryo-EM or ET) of the isolated manchette to prove its intactness.

Line 81: Myosin -> myosin (to be consistent with other protein names)

This work is a significant step toward the understanding of manchettes. While the molecular assignment of dynein and

ATPase is not fully decisive, due to limitation of resolution (this reviewer thinks the assignment of actin filament is convincing, based on its helical symmetry), their speculative model still deserves publication.

2. Significance:

Significance (Required)

This work is a significant step toward the understanding of manchettes. While the molecular assignment of dynein and ATPase is not fully decisive, due to limitation of resolution (this reviewer thinks the assignment of actin filament is convincing, based on its helical symmetry), their speculative model still deserves publication.

3. How much time do you estimate the authors will need to complete the suggested revisions:

Estimated time to Complete Revisions (Required)

(Decision Recommendation)

Between 1 and 3 months

4. Review Commons values the work of reviewers and encourages them to get credit for their work. Select 'Yes' below to register your reviewing activity at Web of Science Reviewer Recognition Service (formerly Publons); note that the content of your review will not be visible on Web of Science.

Yes

Review #3

1. Evidence, reproducibility and clarity:

Evidence, reproducibility and clarity (Required)

****Summary:****

The manchette is a temporary microtubule (MT)-based structure essential for the development of the highly polarised sperm cell. In this study, the authors employed cryo-electron tomography (cryo-ET) and proteomics to investigate the intra-manchette transport system. Cryo-EM analysis of purified rat manchette revealed a high density of MTs interspersed with actin filaments, which appeared either bundled or as single filaments. Vesicles were observed among the MTs, connected by stick-like densities that, based on their orientation relative to MT polarity, were inferred to be kinesins. Subtomogram averaging (STA) confirmed the presence of dynein motor proteins. Proteomic analysis further validated the presence of dynein and kinesins and showed the presence of actin crosslinkers that could bundle actin filaments. Proteomics data also indicated the involvement of actin-based transport mediated by myosin. Importantly, the data indicated that the intraflagellar transport (IFT) system is not part of the intra-manchette transport mechanism. The visualisation of motor proteins directly from a biological sample represents a notable technical advancement, providing new insights into the organisation of the intra-manchette transport system in developing sperm.

Are the key conclusions convincing?

Below we comment on three main conclusions.

MT and F-actin bundles are both constituents of the manchette

While the data convincingly shows that MT and F-actin are part of the manchette, one cannot conclude from it that F-actin is an integral part of the manchette. The authors would need to rephrase so that it is clear that they are speculating.

The transport system employs different transport machinery on these MTs

Proteomics data indicates the presence of multiple motor proteins in the manchette, while cryo-EM data corroborates

this by revealing morphologically distinct densities associated with the MTs. However, the nature of only one of these MT-associated densities has been confirmed-specifically, dynein, as identified through STA. The presence of kinesin or myosin in the EM data remains unconfirmed based on just the cryo-ET density, and therefore it is unclear whether these proteins are actively involved in cargo transport, as this cannot be supported by just the proteomics data. In summary, we recommend that the authors rephrase this conclusion and avoid using the term "employ".

Dynein mediated transport (Line 225-227)

The data shows that dynein is present in the manchette; however, whether it plays an active role in transport cannot be determined from the cryo-ET data provided in the manuscript, as it does not clearly display a dynein-dynactin complex attached to cargo. The attachment to cargo is also not revealed via proteomics as no adaptor proteins that link dynein-dynactin to its cargo have been shown.

Should the authors qualify some of their claims as preliminary or speculative, or remove them altogether?

F-actin

- In the abstract, the authors state that F-actin provides tracks for transport as well as having structural and mechanical roles. However, the manuscript does not include experiments demonstrating a mechanical role. The authors appear to base this statement on literature where actin bundles have been shown to play a mechanical role in other model systems. We suggest they clarify that the mechanical role the authors suggest is speculative and add references if appropriate.

- Lines 15,92, 180 and 255: The statement "Filamentous actin is an integral part of the manchette" is misleading. While the authors show that F-actin is present in their purified manchette structures, whether it is integral has not been tested. Authors should rephrase the sentence.

- To support the claim that F-actin plays a role in transport within the manchette, the authors present only one instance where an unidentified density is attached to an actin filament. This is insufficient evidence to claim that it is myosin actively transporting cargo. Although the proteomics data show the presence of myosin, we suggest the authors exercise more caution with this claim.

- The authors mention the presence of F-actin bundles but do not show direct crosslinking between the F-actin filaments. They could in principle just be closely packed F-actin filaments that are not necessarily linked, so the term "bundle" should be used more cautiously.

Observations of dynein

- Relating to Figure 2B: From the provided image it is not clear whether the density corresponds to a dynein complex, as it does not exhibit the characteristic morphological features of dynein or dynactin molecules.

- Lines 171-172 and Figure 4: It is well established that dynein is a dimer and should always possess two motor domains. The authors have incorrectly assumed they observed single motor heads, except possibly in Figure 4A (marked by an arrow). In all other instances, the dynein complexes show two motor domains in proximity, but these have not been segmented accurately. Furthermore, the "cargos" shown in grey are more likely to represent dynein tails or the dynactin molecule, based on comparisons with in vitro structures of these complexes (see references 1-3).

- Lines 21, 173, and 233 mention cargos, but as noted above, it seems to be parts of the dynein complex the authors are referring to.

- Panel 4B appears to show a dynein-dynactin complex, but whether there is a cargo is unclear and if there is it should be labelled accordingly. To assess whether there is any cargo bound to the dynein-dynactin complex a larger crop of the panel would be helpful.

In summary, we recommend that the authors revisit their segmentations in Figures 2B and 4, revise their text based on these observations, and perform quantification of the data (as suggested in the next section).

Dynein versus kinesin-based transport

The calculation presented in lines 147-151 does not account for the fact that both the dynein-dynactin complex and kinesin proteins require cargo adaptors to transport cargo. Additionally, the authors overlook the possibility that multiple motors could be attached to a single cargo. If the authors did not observe this, they should explicitly mention it to support their argument. In short, the calculations are based on an incorrect premise, rendering the comparison inaccurate. Unless the authors have identified any dynein-dynactin or kinesin cargo adaptors in their proteomics data which could be used for such a comparison, we believe the authors lack sufficient data to accurately estimate the

"active transport ratio" between dynein and kinesin.

Would additional experiments be essential to support the claims of the paper?

F-actin distance and length distribution

- To support the claim that F-actin is bundled (line 189), could the authors provide the distance between each F-actin filament and its neighbours? Additionally, could they compare the average distance to the length of actin crosslinkers found in their proteomics data, or compare it to the distances between crosslinked F-actin observed in other research studies?
- While showing that F-actin is important for the manchette would require cellular experiments, authors could provide quantification of how frequently these actin structures are observed in comparison to MTs to support their claims that these actin filaments could be important for the manchette structure.
- In line 193, the authors claim that the F-actin in bundles appears too short for transport. Could they provide length distributions for these filaments? This might provide further support to their claim that individual F-actin filaments can serve as transport tracks (line 266).
- Could the authors also quantify the abundance of individual F-actin filaments observed, compared to MTs and F-actin bundles, to support the idea that they could play a role in transport?
- In the discussion, the authors mention "interactions between F-actin singlets and mMTs" (line 269), yet they report observing only one instance of this interaction (lines 210 and 211). Given the limited data, they should refer to this as a single interaction in the discussion. The scarcity of data raises questions about how representative this event truly is.

Quantifications for judgement of representativity

The authors should quantify how often they observed vesicles with a stick-like connection to MTs (lines 106-107); this would strengthen the interpretation of the density, as currently only one example is shown in the manuscript (Figure 4A). If possible, they could show how many of them are facing towards the MT plus end.

Dynein quantifications

- The authors are recommended to quantify how many dynein molecules per micron of MT they observe and how often they are angled with their MT binding domain towards the minus-end.
- Could the authors quantify how many dynein densities they found to be attached to a (vesicle) cargo, if any (line 175)? They could show these observations in a supplementary figure.
- For densities that match the size and location of dynein but lack clear dynein morphology (as seen in Figure 2B), could the authors quantify how many are oriented towards the MT minus end?

Artefacts due to purification: Authors should discuss if the purification could have effects on visualizing components of the manchette. For example, if it has effect on the MTs and actin structure or the abundance/structure of the motor protein complexes (bound to cargo or isolated).

Are the experiments adequately replicated and statistical analysis adequate?

The cryo-ET data presented in the manuscript is collected using two separate sample preparations. Along with the quantifications of the different observations suggested above which will help the reader assess how abundant and representative these observations are, the authors could further strengthen their claims by acquiring data from a third sample preparation and then analysing how consistent their observations are between different purifications. This however could be time consuming so it is not a major requirement but recommended if possible within a short time frame.

Are the suggested experiments realistic in terms of time and resources? It would help if you could add an estimated cost and time investment for substantial experiments.

Most of the comments deal with either modifying the text or analysing the data already presented, so the revision could be done with 1-3 months.

****Minor comments:****

Specific experimental issues that are easily addressable.

1. Could the authors state how many tilt series were collected for each dataset/independent sample preparation? We recommend that they upload their raw data or tomograms to EMPAIR.
2. It is not clear to me if the same sample was used for cryo-ET and proteomics. Could the authors clarify how comparable the sample preparation for the cryo-ET and proteomics data is or if the same sample was used for both. If there is a discrepancy between these preparations, they would need to discuss how this can affect comparing observations from cryo-ET and mass spectrometry. Ideally both samples should be the same.

Are prior studies referenced appropriately?

We recommend including additional references to support the claim that F-actin has a mechanical role (line 242). Could the authors compare their proteomics data to other mass spectrometry studies conducted on the Manchette (for example see reference 4)?

Are the text and figures clear and accurate?

Text: We do not see the necessity of specifying the microtubules (MTs) in the data as "manchette MTs" or "mMTs" rather than simply "MTs". However, we recommend that the authors use either "MT" or "mMT" consistently throughout the manuscript.

The authors appear to refer to both dynein-1 (cytoplasmic dynein) and dynein-2 (axonemal dynein or IFT dynein). To avoid confusion, it is important that the authors clearly specify which dynein they are referring to throughout the text. This is particularly relevant as the study aims to demonstrate that IFT is not part of the manchette transport system.

- Introduction: In the third paragraph (lines 59-75), the authors should specify that they are referring to dynein-2, which is distinct from cytoplasmic dynein discussed in the previous paragraph (lines 44-58).
- Figure 4D: The authors could fit a dynein-1 motor domain instead of a dynein-2 into the density to stay consistent with the fact that the density belongs to cytoplasmic dynein-1.

Figures:

- Figure 2B: The legend mentions a large linker complex; however, this may correspond to two or three separate densities.
- Figure 4: please revisit the segmentation of this whole figure based on previous comments.
- Figures 1, 2, 4, 5, and 6: It would be helpful to state in the legends that the tomograms are denoised. There are stripe-like densities visible in the images (e.g., in the vesicle in Figure 2B). Do these artefacts also appear in the raw data?

Do you have suggestions that would help the authors improve the presentation of their data and conclusions?

We suggest revising the paragraph title "Dynein-mediated cargo along the manchette" (line 165) to "Dynein-mediated cargo transport along the manchette".

We recommend that the authors provide additional evidence to support the interpretation that the observed EM densities correspond to motor proteins. Specifically:

- Include scale bars or reference lines indicating the known dimensions of motor proteins, based on previous data, to demonstrate that the observed densities match the expected size.
- Make direct comparisons to existing EM data and highlight morphological similarities.

In the discussion (lines 249-254), the authors could speculate on alternative roles for the IFT components in the manchette, particularly if they are not part of the IFT trains.

We also suggest rephrasing the claim in line 266 to make it more speculative in tone.

Finally, a schematic overview of the manchette ultrastructure in a spermatid would greatly aid the reader in understanding the material presented.

****References:****

1. Chowdhury, S., Ketcham, S., Schroer, T. et al. Structural organization of the dynein-dynactin complex bound to microtubules. *Nat Struct Mol Biol* 22, 345-347 (2015). <https://doi.org/10.1038/nsmb.2996>
2. Grotjahn, D.A., Chowdhury, S., Xu, Y. et al. Cryo-electron tomography reveals that dynactin recruits a team of dyneins for processive motility. *Nat Struct Mol Biol* 25, 203-207 (2018). <https://doi.org/10.1038/s41594-018-0027-7>
3. Chaaban, S., Carter, A.P. Structure of dynein-dynactin on microtubules shows tandem adaptor binding. *Nature* 610, 212-216 (2022). <https://doi.org/10.1038/s41586-022-05186-y>
4. W. Hu, R. Zhang, H. Xu, Y. Li, X. Yang, Z. Zhou, X. Huang, Y. Wang, W. Ji, F. Gao, W. Meng, CAMSAP1 role in orchestrating structure and dynamics of manchette microtubule minus-ends impacts male fertility during spermiogenesis, *Proc. Natl. Acad. Sci. U.S.A.* 120 (45) e2313787120, <https://doi.org/10.1073/pnas.2313787120> (2023).

2. Significance:

Significance (Required)

This study employs cryo-electron tomography (cryo-ET) and proteomics to elucidate the architecture of the manchette. It advances our understanding of the components involved in intracellular transport within the manchette and introduces the following technical and conceptual innovations:

a) *Technical Advances:*

The authors have visualized the manchette at high resolution using cryo-ET. They optimized a purification pipeline capable of retaining, at least partially, the transport machinery of the manchette. Notably, they observed dynein and putative kinesin motors attached to microtubules—a significant achievement that, to our knowledge, has not been reported previously.

b) *Conceptual Advances:*

This study provides novel insights into spermatogenesis. The findings suggest that intraflagellar transport (IFT) is unlikely to play a role at this stage of sperm development while shedding light on alternative transport systems. Importantly, the authors demonstrate that actin filaments organize in two distinct ways: clustering parallel to microtubules or forming single filaments.

This work is likely to be of considerable interest to researchers in sperm development and structural biology. Additionally, it may appeal to scientists studying motor proteins and the cytoskeleton.

The reviewers possess extensive expertise in in situ cryo-electron tomography and single-particle microscopy, including work on dynein-based complexes. Collectively, they have significant experience in the field of cytoskeleton-based transport.

3. How much time do you estimate the authors will need to complete the suggested revisions:

Estimated time to Complete Revisions (Required)

(Decision Recommendation)

Between 1 and 3 months

4. Review Commons values the work of reviewers and encourages them to get credit for their work. Select 'Yes' below to register your reviewing activity at Web of Science Reviewer Recognition Service (formerly Publons); note that the content of your review will not be visible on Web of Science.

Yes

Reviewer #1 (Evidence, reproducibility and clarity (Required)):

In this manuscript the authors have done cryo-electron tomography of the manchette, a microtubule-based structure important for proper sperm head formation during spermatogenesis. They also did mass-spectrometry of the isolated structures. Vesicles, actin and their linkers to microtubules within the structure are shown.

We thank the reviewer for the critical reading of our manuscript; we have implemented the suggestions as detailed below, which we believe indeed improved the manuscript.

Major:

The data the conclusions are based on seem very limited and sometimes overinterpreted. For example, only one connection between actin and microtubules was observed, and this is thought to be MACF1 simply based on its presence in the MS.

We regret giving the impression that the data is limited. We in fact collected >100 tilt series from 3 biological replicas for the isolated manchette.

In the revised version, we added data from in-situ studies showing vesicles interacting with the manchette (as requested below, new Fig. 1).

Specifically, for the interaction of actin with microtubule we added more examples (Revised Fig. 6) and we toned down the discussion related to the relevance of this interaction (lines 193-194, 253-255). MACF1 is mentioned only as a possible candidate in the discussion (line 254).

Another, and larger concern, is that the authors do a structural study on something that has been purified out of the cell, a process which is extremely disruptive. Vesicles, actin and other cellular components could easily be trapped in this cytoskeletal sieve during the purification process and as such, not be bona fide manchette components. This could create both misleading proteomics and imaging. Therefore, an approach not requiring extraction such as high-pressure freezing, sectioning and room-temperature electron tomography and/or immunoEM on sections to set aside this concern is strongly recommended. As an additional bonus, it would show if the vesicles containing ATP synthase are deformed mitochondria.

We recognise the concern raised by the reviewer.

To alleviate this concern, we added imaging data of manchettes in-situ that show vesicles, mitochondria and filaments interacting with the manchette (new Fig. 1), essentially confirming the observations that were made on the isolated manchette.

The benefits of imaging the isolated manchette were better throughput (being able to collect more data) and reaching higher resolution allowing to resolve unequivocally the dynein/dynactin and actin filaments.

Minor:

Line 99: "to study IMT with cryo-ET, manchettes were isolated ...(insert from which organism)..."
Added in line 102 in the revised version.

Line 102 "...demonstrating that they can be used to study IMT".. can the authors please clarify?
This paragraph was revised (lines 131-137), we hope it is now more clear.

Line 111 "densities face towards the MT plus-end" How can a density "face" anywhere? For this, it needs to have a defined front and back.

Microtubule motor proteins (kinesin and dynein) are often attached to the microtubules with an angle and dynactin and cargo on one side (plus end). We rephrased this part and removed the word "face" in the revised version to make it more clear (lines 161-162).

Line 137: is the "perinuclear ring" the same as the manchette?

The perinuclear ring is the apical part of the manchette that connects it to the nucleus. We added to the revised version imaging of the perinuclear ring with observations on how it changes when the manchette elongates (new Fig. 2).

Figure 2B: How did the authors decide not to model the electron density found between the vesicle and the MT at 3 O'clock? Is there no other proteins with a similar lollipop structure as ATP synthase, so that this can be said to be this protein with such certainty?

The densities connecting the vesicles to the microtubules shown in (now) Fig. 4D are not consistent enough to be averaged.

The densities resembling ATP synthase are inside the vesicles. Nevertheless, we have decided to remove the averaging of the ATP synthases from the revised manuscript as they are not of great importance for this manuscript. Instead, the new in-situ data clearly show mitochondria (with their characteristic double membrane and cristae) interacting with manchette microtubule (new Fig 1C).

Line 189: "F-actin formed organized bundles running parallel to mMTs" - this observation needs confirming in a less disrupted sample.

Phalloidin (actin marker) was shown before to stain the manchette (PMID: 36734600). As actin filaments are very thin (7 nm) they are very hard to observe in plastic embedded EM.

In the in-situ data we added to the revised manuscript (new Fig 1D), we observe filaments with a diameter corresponding to actin. In addition, we added more examples of microtubules interacting with actin in isolated manchette (new Fig. 6 E-K).

Line 242 remove first comma sign

Removed.

Line 363 "a total of 2 datasets" - is this manuscript based on only two tilt-series? Or two datasets from each of the 4 grids? In any case, this is very limited data.

We apologise for not clearly providing the information about the data size in the original manuscript. The data is based on three biological replicas (3 animals). We collected more than 100 tomograms of different regions of the manchettes. As such, we would argue that the data is not limited per se.

Reviewer #1 (Significance (Required)):

The article is very interesting, and if presented together with the suggested controls, would be informative to both microtubule/motorprotein researchers as well as those trying studying spermatogenesis.

Reviewer #2 (Evidence, reproducibility and clarity (Required)):

The manchette appears as a shield-like structure surrounding the flagellar basal body upon spermiogenesis. It consists of a number of microtubules like a comb, but actin (Mochida et al. 1998 Dev. Biol. 200, 46) and myosin (Hayasaka et al. 2008 Asian J. Androl. 10, 561) were found, suggesting transportation inside the manchette. Detailed structural information and functional insight into the manchette was still awaited. There is a hypothesis called IMT (intra-machette transport) based on the fact that manchette and IFT (intraflagellar transport) share common components (or homologues) and on their transition along the stages of spermiogenesis. While IMT is considered as a potential hypothesis to explain delivery of centrosomal and flagellar components, no one has witnessed IMT at the same level as IFT. IMT has never been purified, visualized in motion or at high resolution.

This study for the first time visualized manchette using high-end cryo-electron tomography of isolated manchettes, addressing structural characterization of IMT. The authors successfully microtubular bundles, vesicles located between microtubules and a linker-like structure connecting the vesicle and the microtubule. On multilamellar membranes in the vesicles they found particles and assigned them to ATPase complexes, based on intermediate (~60Å) resolution structure. They further identified interesting structures, such as (1) particles on microtubules, which resemble dynein and (2) filaments which shows symmetry of F-actin. All the molecular assignments are consistent with their proteomics of manchettes.

We thank the reviewer for highlighting the novelty of our study.

Their assignment of ATPase will be strengthened by MS data, if it proves absence of other possible proteins forming such a membrane protein complex.

All the ATPase components were indeed found in our proteomics data. Nevertheless, we have decided to remove the averaging of the ATPase as it does not directly relate to IMT, the focus of this manuscript.

They discussed possible role of various motor proteins based on their abundance (Line 134-151, Line 200). This makes sense only with a control. Absolute abundance of proteins would not necessarily present their local importance or roles. This reviewer would suggest quantitative proteomics of other organelles, or whole cells, or other fractions obtained during manchette isolation, to demonstrate unique abundance of KIF27 and other proteins of their interest.

We agree with the reviewer that absolute abundance does not necessarily indicate importance or a role. As such, we removed this part of the discussion from the revised manuscript.

A single image from a tomogram, Fig.6B, is not enough to prove actin-MT interaction. A gallery and a number (how many such junctions were found from how many MTs) will be necessary.

We agree that one example is not enough. In the new Fig. 6E-K, we provide a gallery of more examples. We have revised the text to reflect the point that these observations are still rare and more data will be needed to quantify this interaction (Lines 253-254).

Minor points:

Their manchette purification is based on Mochida et al., which showed (their Fig.2) similarity to the in vivo structure (for example, Fig.1 of Kierszenbaum 2001 Mol. Reproduc. Dev. 59, 347).

Nevertheless, since this is not a very common prep, it is helpful to show the isolated manchette's wide view (low mag cryo-EM or ET) to prove its intactness.

We thank the reviewer for this suggestion, in the revised version, new Fig. 2 provides a cryo-EM overview of purified manchette from different developmental stages.

Line 81: Myosin -> myosin (to be consistent with other protein names)

Corrected.

This work is a significant step toward the understanding of manchettes. While the molecular assignment of dynein and ATPase is not fully decisive, due to limitation of resolution (this reviewer thinks the assignment of actin filament is convincing, based on its helical symmetry), their speculative model still deserves publication.

Reviewer #2 (Significance (Required)):

This work is a significant step toward the understanding of manchettes. While the molecular assignment of dynein and ATPase is not fully decisive, due to limitation of resolution (this reviewer thinks the assignment of actin filament is convincing, based on its helical symmetry), their speculative model still deserves publication.

Reviewer #3 (Evidence, reproducibility and clarity (Required)):

->Summary:

The manchette is a temporary microtubule (MT)-based structure essential for the development of the highly polarised sperm cell. In this study, the authors employed cryo-electron tomography (cryo-ET) and proteomics to investigate the intra-manchette transport system. Cryo-EM analysis of purified rat manchette revealed a high density of MTs interspersed with actin filaments, which appeared either bundled or as single filaments. Vesicles were observed among the MTs, connected by stick-like densities that, based on their orientation relative to MT polarity, were inferred to be kinesins. Subtomogram averaging (STA) confirmed the presence of dynein motor proteins. Proteomic analysis further validated the presence of dynein and kinesins and showed the presence of actin crosslinkers that could bundle actin filaments. Proteomics data also indicated the involvement of actin-based transport mediated by myosin. Importantly, the data indicated that the intraflagellar transport (IFT) system is not part of the intra-manchette transport mechanism. The visualisation of motor proteins directly from a biological sample represents a notable technical advancement, providing new insights into the organisation of the intra-manchette transport system in developing sperm.

We thank the reviewer for summarising the novelty of our observations.

-> Are the key conclusions convincing?

Below we comment on three main conclusions.

MT and F-actin bundles are both constituents of the manchette

While the data convincingly shows that MT and F-actin are part of the manchette, one cannot conclude from it that F-actin is an integral part of the manchette. The authors would need to rephrase so that it is clear that they are speculating.

We have rephrased our statements and replaced “integral” with ‘actin filaments are associated’. Of note previous studies suggested actin are part of the manchette including staining with phalloidin (PMID: 36734600, PMID: 9698455, PMID: 18478159) and we here visualised the actin in high resolution.

The transport system employs different transport machinery on these MTs

Proteomics data indicates the presence of multiple motor proteins in the manchette, while cryo-EM data corroborates this by revealing morphologically distinct densities associated with the MTs.

However, the nature of only one of these MT-associated densities has been confirmed-specifically, dynein, as identified through STA. The presence of kinesin or myosin in the EM data remains unconfirmed based on just the cryo-ET density, and therefore it is unclear whether these proteins are actively involved in cargo transport, as this cannot be supported by just the proteomics data. In summary, we recommend that the authors rephrase this conclusion and avoid using the term "employ".

We agree that our cryo-ET only confirmed the motor protein dynein. As such, we removed the term employ and rephrased our claims regarding the active transport and accordingly changed the title.

Dynein mediated transport (Line 225-227)

The data shows that dynein is present in the manchette; however, whether it plays an active role in transport cannot be determined from the cryo-ET data provided in the manuscript, as it does not clearly display a dynein-dynactin complex attached to cargo. The attachment to cargo is also not revealed via proteomics as no adaptor proteins that link dynein-dynactin to its cargo have been shown.

A list of cargo adaptor proteins were found in our proteomics data but we agree that cryo-ET and proteomics alone cannot prove active transport. As such we toned down the discussion about active transport (lines 212-220).

-> Should the authors qualify some of their claims as preliminary or speculative, or remove them altogether?

F-actin

- In the abstract, the authors state that F-actin provides tracks for transport as well as having structural and mechanical roles. However, the manuscript does not include experiments demonstrating a mechanical role. The authors appear to base this statement on literature where actin bundles have been shown to play a mechanical role in other model systems. We suggest they clarify that the mechanical role the authors suggest is speculative and add references if appropriate.

We removed the claim about the mechanical role of the actin from the abstract and rephrased this in the discussion to suggest this role for the F-actin (lines 242-243).

- Lines 15,92, 180 and 255: The statement "Filamentous actin is an integral part of the manchette" is misleading. While the authors show that F-actin is present in their purified manchette structures, whether it is integral has not been tested. Authors should rephrase the sentence.

We removed the word integral.

- To support the claim that F-actin plays a role in transport within the manchette, the authors present only one instance where an unidentified density is attached to an actin filament. This is insufficient evidence to claim that it is myosin actively transporting cargo. Although the proteomics data show the presence of myosin, we suggest the authors exercise more caution with this claim.

We agree that our data do not demonstrate active transport as such we removed that claim. We mention the possibility of cargo transport in the discussion (lines 250-255).

- The authors mention the presence of F-actin bundles but do not show direct crosslinking between the F-actin filaments. They could in principle just be closely packed F-actin filaments that are not necessarily linked, so the term "bundle" should be used more cautiously.

We do not assume that a bundle means that the F-actin filaments are crosslinked. A bundle simply indicates the presence of multiple F-actin filaments together. We rephrased it to call them actin clusters.

Observations of dynein

- Relating to Figure 2B: From the provided image it is not clear whether the density corresponds to a dynein complex, as it does not exhibit the characteristic morphological features of dynein or dynactin molecules.

We indeed do not claim that the densities in this figure are dynein or dynactin. We revised this paragraph and hope that it is now more clear (lines 135-137).

- Lines 171-172 and Figure 4: It is well established that dynein is a dimer and should always possess two motor domains. The authors have incorrectly assumed they observed single motor heads, except possibly in Figure 4A (marked by an arrow). In all other instances, the dynein complexes show two motor domains in proximity, but these have not been segmented accurately. Furthermore, the "cargos" shown in grey are more likely to represent dynein tails or the dynactin molecule, based on comparisons with in vitro structures of these complexes (see references 1-3).

We thank the reviewer for this correction. We improved the annotations in the figure and revised the text to clarify that we identified dimers of dynein motor heads (lines 140-144). We further added a projection of a dynein dynactin complex to compare to the observation on the manchette (new Fig. 5E). We further changed claims on the presence of protein cargo to the presence of dynein/dynactin that allows cargo tethering based on the presence of cargo adaptors in the proteomics data.

- Lines 21, 173, and 233 mention cargos, but as noted above, it seems to be parts of the dynein complex the authors are referring to.

This was corrected as mentioned above.

- Panel 4B appears to show a dynein-dynactin complex, but whether there is a cargo is unclear and if there is it should be labelled accordingly. To assess whether there is any cargo bound to the dynein-dynactin complex a larger crop of the panel would be helpful

In summary, we recommend that the authors revisit their segmentations in Figures 2B and 4, revise their text based on these observations, and perform quantification of the data (as suggested in the next section).

We thank the reviewers for sharing their expertise on dynein-dynactin complexes. We have revised the text as detailed above and excluded the assignment of any cargo, as we cannot (even from larger panels) see a clear association of cargo. We have made clear that we only refer to dynein dynactin with the capability of linking cargo based on the presence of proteomics data. We have removed claims on active transport with dynein.

Dynein versus kinesin-based transport

The calculation presented in lines 147-151 does not account for the fact that both the dynein-dynactin complex and kinesin proteins require cargo adaptors to transport cargo. Additionally, the authors overlook the possibility that multiple motors could be attached to a single cargo. If the authors did not observe this, they should explicitly mention it to support their argument. In short, the calculations are based on an incorrect premise, rendering the comparison inaccurate. Unless the authors have identified any dynein-dynactin or kinesin cargo adaptors in their proteomics data which could be used for such a comparison, we believe the authors lack sufficient data to accurately estimate the "active transport ratio" between dynein and kinesin.

Even though we detect cargo adaptors in our proteomics, we agree that calculating relative transport based only on the proteomics can be inaccurate as such we removed absolute quantification and comparison between dynein and kinesin-based IMT.

- Would additional experiments be essential to support the claims of the paper?

F-actin distance and length distribution

- To support the claim that F-actin is bundled (line 189), could the authors provide the distance between each F-actin filament and its neighbours? Additionally, could they compare the average distance to the length of actin crosslinkers found in their proteomics data, or compare it to the distances between crosslinked F-actin observed in other research studies?

We measured distances between the actin filaments and added a plot to new Fig 6.

- While showing that F-actin is important for the manchette would require cellular experiments, authors could provide quantification of how frequently these actin structures are observed in comparison to MTs to support their claims that these actin filaments could be important for the manchette structure.

We agree that claims on the role and function of actin in the manchette require cellular experiments that are beyond the scope of this study. Absolute quantification of the ratio between MTs and actin from cryoET is very hard and will be inaccurate as the manchette cannot be imaged as a whole due to its size and thickness. The ratio we have is based on the relative abundance provided by the proteomics (Fig. 5F).

- In line 193, the authors claim that the F-actin in bundles appears too short for transport. Could they provide length distributions for these filaments? This might provide further support to their claim that individual F-actin filaments can serve as transport tracks (line 266).

In addition to the limitation mentioned in the previous point, quantification of length from high magnification imaging will likely be inaccurate as the length of the actin in most cases is bigger than the field of view that is captured. Nevertheless, we removed the claim about the actin being too short for transport.

- Could the authors also quantify the abundance of individual F-actin filaments observed, compared to MTs and F-actin bundles, to support the idea that they could play a role in transport?

As explained for the above points absolute quantification of the ratio between MTs and actin is not feasible from cryoET data that cannot capture all of the manchette in high enough resolution to resolve the actin.

- In the discussion, the authors mention "interactions between F-actin singlets and mMTs" (line 269), yet they report observing only one instance of this interaction (lines 210 and 211). Given the limited data, they should refer to this as a single interaction in the discussion. The scarcity of data raises questions about how representative this event truly is.

We agree that one example is not enough. In the new Fig. 6E-K, we provide a gallery of more examples as also requested by reviewers 1 and 2. We have also revised the text to reflect the point that these observations are still rare (Lines 190-194).

Quantifications for judgement of representativity

The authors should quantify how often they observed vesicles with a stick-like connection to MTs (lines 106-107); this would strengthen the interpretation of the density, as currently only one example is shown in the manuscript (Figure 4A). If possible, they could show how many of them are facing towards the MT plus end.

As mentioned in the text (lines 135-137), the linkers connecting vesicles to MTs were irregular and so we could not interpret them further this is in contrast to dynein that were easily recognisable but were not associated with vesicles.

Dynein quantifications

- The authors are recommended to quantify how many dynein molecules per micron of MT they observe and how often they are angled with their MT binding domain towards the minus-end.

As the manchette is large and highly dense any quantification will likely be biased towards parts of the manchette that are easier to image, for example the periphery. As such we do not think quantifying the dynein density will yield meaningful insight.

- Could the authors quantify how many dynein densities they found to be attached to a (vesicle) cargo, if any (line 175)? They could show these observations in a supplementary figure.

We did not observe any case of a connection between a vesicle and dynein motors, we edited this sentence to be more clear on that.

- For densities that match the size and location of dynein but lack clear dynein morphology (as seen in Figure 2B), could the authors quantify how many are oriented towards the MT minus end?

We had many cases where the connection did not have a clear dynein morphology, and as the morphology is not clear, it is impossible to make a claim about whether they are oriented towards the minus end.

Artefacts due to purification: Authors should discuss if the purification could have effects on visualizing components of the manchette. For example, if it has effect on the MTs and actin structure or the abundance/structure of the motor protein complexes (bound to cargo or isolated).

We have followed a protocol that was published before and showed the overall integrity of the manchette. Nevertheless, losing connections between manchette and other cellular organelles are expected. To address this point, we added in-situ data (new Fig 1) showing manchette in intact spermatids interacting with vesicles and mitochondria, as well as overviews of manchettes (new Fig 2), the text was revised accordingly.

- Are the experiments adequately replicated and statistical analysis adequate?

The cryo-ET data presented in the manuscript is collected using two separate sample preparations.

Along with the quantifications of the different observations suggested above which will help the reader assess how abundant and representative these observations are, the authors could further strengthen their claims by acquiring data from a third sample preparation and then analysing how consistent their observations are between different purifications. This however could be time consuming so it is not a major requirement but recommended if possible within a short time frame. **We regret not explicitly mentioning our data set size, it was added now to the revised version. In essence, the data is based on three biological replicas (3 animals). We collected more than 100 tomograms of different regions of the manchettes. We provided in the revised version more observations (new Fig 1, 2, 4B-C and 6E-K).**

- Are the suggested experiments realistic in terms of time and resources? It would help if you could add an estimated cost and time investment for substantial experiments.

Most of the comments deal with either modifying the text or analysing the data already presented, so the revision could be done with 1-3 months.

Minor comments:

- Specific experimental issues that are easily addressable.

1) Could the authors state how many tilt series were collected for each dataset/independent sample preparation? We recommend that they upload their raw data or tomograms to EMPAIR.

We added this information in the material and methods.

2) It is not clear to me if the same sample was used for cryo-ET and proteomics. Could the authors clarify how comparable the sample preparation for the cryo-ET and proteomics data is or if the same sample was used for both. If there is a discrepancy between these preparations, they would need to discuss how this can affect comparing observations from cryo-ET and mass spectrometry. Ideally both samples should be the same.

After sample preparation the manchettes were directly frozen on grids. The rest of the samples was used for proteomics. Consequently, EM and MS data were acquired on the same samples. We clarified this in the text (lines 327-328).

- Are prior studies referenced appropriately?

We recommend including additional references to support the claim that F-actin has a mechanical role (line 242).

Could the authors compare their proteomics data to other mass spectrometry studies conducted on the Manchette (for example, see reference 4)?

We added the comparison but it is important to point out that in reference 4 the manchettes were isolated from mice testes.

- Are the text and figures clear and accurate?

Text: We do not see the necessity of specifying the microtubules (MTs) in the data as "manchette MTs" or "mMTs" rather than simply "MTs". However, we recommend that the authors use either "MT" or "mMT" consistently throughout the manuscript.

We changed to only MTs.

The authors appear to refer to both dynein-1 (cytoplasmic dynein) and dynein-2 (axonemal dynein or IFT dynein). To avoid confusion, it is important that the authors clearly specify which dynein they are referring to throughout the text. This is particularly relevant as the study aims to demonstrate that IFT is not part of the manchette transport system.

- Introduction: In the third paragraph (lines 59-75), the authors should specify that they are referring

to dynein-2, which is distinct from cytoplasmic dynein discussed in the previous paragraph (lines 44-58).

We specify the respective dyneins in the text (line 66,140-141,145).

- Figure 4D: The authors could fit a dynein-1 motor domain instead of a dynein-2 into the density to stay consistent with the fact that the density belongs to cytoplasmic dynein-1.

We changed the figure and fitted a cytosolic dynein-1 structure (5nvu) instead.

Figures:

- Figure 2B: The legend mentions a large linker complex; however, this may correspond to two or three separate densities.

We have addressed this and changed the wording.

- Figure 4: please revisit the segmentation of this whole figure based on previous comments.

We revised as suggested.

- Figures 1, 2, 4, 5, and 6: It would be helpful to state in the legends that the tomograms are denoised. There are stripe-like densities visible in the images (e.g., in the vesicle in Figure 2B). Do these artefacts also appear in the raw data?

As stated in the Methods section, tomograms were generally denoised with CryoCare for visualisation purposes. The “stripe-like densities” are artefacts of the gold fiducials used for tomogram alignment and appear in the raw data (before denoising).

- Do you have suggestions that would help the authors improve the presentation of their data and conclusions?

We suggest revising the paragraph title "Dynein-mediated cargo along the manchette" (line 165) to "Dynein-mediated cargo transport along the manchette".

We have changed this in the revised version.

We recommend that the authors provide additional evidence to support the interpretation that the observed EM densities correspond to motor proteins. Specifically:

- Include scale bars or reference lines indicating the known dimensions of motor proteins, based on previous data, to demonstrate that the observed densities match the expected size.

The dynein structure is provided for reference. We also added the cytosolic dynein–dynactin as a reference (Fig 5E).

- Make direct comparisons to existing EM data and highlight morphological similarities.

We have added a comparison to existing data (Fig 5E).

In the discussion (lines 249-254), the authors could speculate on alternative roles for the IFT components in the manchette, particularly if they are not part of the IFT trains.

We also suggest rephrasing the claim in line 266 to make it more speculative in tone.

We have addressed this in the revised version (lines 221-230).

Finally, a schematic overview of the manchette ultrastructure in a spermatid would greatly aid the reader in understanding the material presented.

We now include a graphical abstract and overviews of isolated manchettes on cryo-EM grids.

References:

1. Chowdhury, S., Ketcham, S., Schroer, T. et al. Structural organization of the dynein-dynactin complex bound to microtubules. *Nat Struct Mol Biol* 22, 345-347 (2015). <https://doi.org/10.1038/nsmb.2996>

2. Grotjahn, D.A., Chowdhury, S., Xu, Y. et al. Cryo-electron tomography reveals that dynactin recruits a team of dyneins for processive motility. *Nat Struct Mol Biol* 25, 203-207 (2018). <https://doi.org/10.1038/s41594-018-0027-7>

3. Chaaban, S., Carter, A.P. Structure of dynein-dynactin on microtubules shows tandem adaptor binding. *Nature* 610, 212-216 (2022). <https://doi.org/10.1038/s41586-022-05186-y>

4. W. Hu, R. Zhang, H. Xu, Y. Li, X. Yang, Z. Zhou, X. Huang, Y. Wang, W. Ji, F. Gao, W. Meng, CAMSAP1 role in orchestrating structure and dynamics of manchette microtubule minus-ends impacts male fertility during spermiogenesis, *Proc. Natl. Acad. Sci. U.S.A.* 120 (45) e2313787120, <https://doi.org/10.1073/pnas.2313787120> (2023).

Reviewer #3 (Significance (Required)):

This study employs cryo-electron tomography (cryo-ET) and proteomics to elucidate the architecture of the manchette. It advances our understanding of the components involved in intracellular transport within the manchette and introduces the following technical and conceptual innovations:

a) Technical Advances:

The authors have visualized the manchette at high resolution using cryo-ET. They optimized a purification pipeline capable of retaining, at least partially, the transport machinery of the manchette. Notably, they observed dynein and putative kinesin motors attached to microtubules—a significant achievement that, to our knowledge, has not been reported previously.

b) Conceptual Advances:

This study provides novel insights into spermatogenesis. The findings suggest that intraflagellar transport (IFT) is unlikely to play a role at this stage of sperm development while shedding light on alternative transport systems. Importantly, the authors demonstrate that actin filaments organize in two distinct ways: clustering parallel to microtubules or forming single filaments.

This work is likely to be of considerable interest to researchers in sperm development and structural biology. Additionally, it may appeal to scientists studying motor proteins and the cytoskeleton.

We thank the reviewers for appreciating the significance and novelty of our study.

The reviewers possess extensive expertise in in situ cryo-electron tomography and single-particle microscopy, including work on dynein-based complexes. Collectively, they have significant experience in the field of cytoskeleton-based transport.

July 8, 2025

RE: Life Science Alliance Manuscript #LSA-2025-03415-T

Dr. Tzviya Zeev-Ben-Mordehai
Utrecht University
Structural Biochemistry, Bijvoet Center for Biomolecular Research
Utrecht, Utrecht 3584 CH
Netherlands

Dear Dr. Zeev-Ben-Mordehai,

Thank you for submitting your revised manuscript entitled "Characterisation of the Spermatid Manchette Architecture and its Role as Transport Scaffold". All reviewers are now satisfied with minor suggestions which we invite you to consider. We would be happy to publish your paper in Life Science Alliance pending final revisions necessary to meet our formatting guidelines.

- Please be sure that the authorship listing and order is correct.
- Please upload your main manuscript text as an editable doc file.
- Please upload all figure files as individual ones, including the supplementary figure files; all figure legends should only appear in the main manuscript file; Graphical Abstract should also be uploaded separately and be labeled as such
- Please upload your Tables in editable .doc or excel format separately.
- Please leave your main, supplementary figure, and table legends in the main manuscript text after the references section.
- Please add a Running Title in our system.
- Please add a Data Availability section, including accession for proteomics data and all supporting data underlying the cryo-ET results.
- Please add ORCID ID for corresponding author--you should have received instructions on how to do so.
- Please add a Category for your manuscript in our system.
- Please add a Summary Blurb/Alternate Abstract in our system.
- Please add the X and Bluesky handles of your host institute/organization as well as your own or/and one of the authors in our system.
- Please use the [10 author names, et al.] format in your references (i.e. limit the author names to the first 10).
- Please add a callouts for Figures 6D, S1A-B, S2A-B and S3B to your main manuscript text.

LSA now encourages authors to provide a 30-60 second video where the study is briefly explained. We will use these videos on social media to promote the published paper and the presenting author (for examples, see <https://docs.google.com/document/d/1-UWCfbE4pGcDdcgzcmiuJl2XMBJnxKYeqRvLLrLSo8s/edit?usp=sharing>). Corresponding or first-authors are welcome to submit the video. Please submit only one video per manuscript. The video can be emailed to contact@life-science-alliance.org

A. FINAL FILES:

-- Summary blurb (enter in submission system): A short text summarizing in a single sentence the study (max. 200 characters including spaces). This text is used in conjunction with the titles of papers, hence should be informative and complementary to the title. It should describe the context and significance of the findings for a general readership; it should be written in the

present tense and refer to the work in the third person. Author names should not be mentioned.

B. MANUSCRIPT ORGANIZATION AND FORMATTING:

Sincerely,

Reviewer #1 (Comments to the Authors (Required)):

The authors addressed my point on reproducibility of actin-MT interaction by providing further images as figure panels. This reviewer found it convincing. For other points by all the reviewers, concerning overstatement, they toned down their conclusion, instead of providing new results. This at least balanced the evidence and the conclusion. This reviewer would accept this manuscript for publication.

Reviewer #2 (Comments to the Authors (Required)):

The authors have made substantial improvements in this revised version of the manuscript. Key sections have been reformulated to ensure that the interpretations and conclusions are more closely aligned with the presented data. The revised manuscript now places greater emphasis on the overall architectural features of the manchette. Notably, the authors have included new in situ evidence demonstrating that vesicles and organelles are located in close proximity to the manchette microtubules, thereby strengthening their claims. Overall, the authors have satisfactorily addressed my previous concerns, and I find this version suitable for publication.

I have a few minor textual suggestions for further clarity and consistency:

Line 56: could benefit from rephrasing. Cargo tethering is done by adaptor proteins which are also required for activation of dynein-dynactin. Consider introducing adaptor proteins more explicitly here, as they are discussed in more detail in the Results and Discussion section.

Line 133: could rephrase to clarify that the in situ data show colocalization, rather than implying it shows physical association.

Line 152: It could be helpful to use "activating adaptors" or simply "adaptors," which are more commonly used terms than "cargo linkers." See: Reck-Peterson et al., 2018.

Lines 174-175: This paragraph discusses bi-directional transport in the absence of IFT, but then refers to "unidirectionality in the manchette" being different from that in the axoneme. This phrasing is unclear. Do the authors mean "microtubule polarity" or "bi-

directional transport mechanisms"? Please clarify. The same issue arises in Line 225.

This paragraph would be a good place to state the kinesins identified in the mass spectrometry data, as they may contribute to plus-end-directed transport along the microtubules.

Line 297: Please confirm whether the unit "500 nA" is accurate. Should this instead be "500 pA"?

Line 681: Consider rephrasing the description of the "dark grey part" to clearly state that it includes both dynein tails and dynactin, for improved clarity.

Reviewer #3 (Comments to the Authors (Required)):

1. This manuscript is the first high-resolution description of the spermatid manchette architecture. The new revised manuscript has increased significantly in impact to the previous version. By adding cryo-ET of FIB-milled samples, my major concerns are completely addressed, in fact they went beyond what I asked for. The way reviewers' concerns have been addressed is in my opinion, exemplary.

2. The main points of the paper are now very well supported in the experimental data.

3. I only have miniscule text edits to suggest:

Line 98: study not studies.

Line 140: exchange regularly to "often" or similar for ease of reading.

Line 183 & 185: two sentences in a row that starts with "while".

Line 195: can the way actin interacts with MTs be more described in the text?

Line 246: 2 times "in the manchette" after each other.

I recommend publication and need not to review it again.

July 21, 2025

RE: Life Science Alliance Manuscript #LSA-2025-03415-TR

Dr. Tzviya Zeev-Ben-Mordehai
Utrecht University
Structural Biochemistry, Bijvoet Center for Biomolecular Research
Utrecht, Utrecht 3584 CH
Netherlands

Dear Dr. Zeev-Ben-Mordehai,

Thank you for submitting your Research Article entitled "Characterisation of the Spermatid Manchette Architecture and its Role as Transport Scaffold". It is a pleasure to let you know that your manuscript is now accepted for publication in Life Science Alliance. Congratulations on this interesting work.

DISTRIBUTION OF MATERIALS:

Again, congratulations on a very nice paper. I hope you found the review process to be constructive and are pleased with how the manuscript was handled editorially. We look forward to future exciting submissions from your lab.

Sincerely,
